# Identification of Key Differentially Expressed Genes in *Arabidopsis thaliana* Under Short- and Long-Term High Light Stress

**DOI:** 10.3390/ijms26167790

**Published:** 2025-08-12

**Authors:** Aleksandr V. Bobrovskikh, Ulyana S. Zubairova, Alexey V. Doroshkov

**Affiliations:** 1The Federal Research Center Institute of Cytology and Genetics, Siberian Branch of the Russian Academy of Sciences, 630090 Novosibirsk, Russia; ulyanochka@bionet.nsc.ru (U.S.Z.); ad@bionet.nsc.ru (A.V.D.); 2Department of Natural Sciences, Novosibirsk State University, 630090 Novosibirsk, Russia; 3Department of Information Technologies, Novosibirsk State University, 630090 Novosibirsk, Russia; 4Department of Genomics and Bioinformatics, Institute of Fundamental Biology and Biotechnology, Siberian Federal University, 660036 Krasnoyarsk, Russia

**Keywords:** plant stress, transcriptomics, meta-analysis, gene regulatory networks, photosynthesis, reactive oxygen species, bioinformatics

## Abstract

Nowadays, with the accumulation of large amounts of stress-response transcriptomic data in plants, it is possible to clarify the key genes and transcription factors (TFs) involved in these processes. Here, we present the comprehensive transcriptomic meta-analysis of the high light (HL) response in photosynthetic tissues of *Arabidopsis thaliana* (L.) Heynh., offering new insights into adaptation mechanisms of plants to excessive light and involved gene regulatory networks. We analyzed 21 experiments covering 58 HL conditions in total, yielding 218,000 instances of differentially expressed genes (DEGs) corresponding to 19,000 unique genes. Based on these data, we developed the publicly accessible AraLightDEGs resource, which offers multiple search filters for experimental conditions and gene characteristics, and we conducted a detailed meta-analysis using our R pipeline, AraLightMeta. Our meta-analysis highlighted distinct transcriptional programs between short- and long-term HL responses in leaves, revealing novel regulatory interactions and refining the understanding of key DEGs. In particular, long-term HL adaptation involves key TFs such as *CRF3* and *PTF1* regulating antioxidant and jasmonate signaling; *ATWHY2*, *WHY3*, and *emb2746* coordinating chloroplast and mitochondrial gene expression; *AT2G28450* governing ribosome biogenesis; and *AT4G12750* controlling methyltransferase activity. We integrated these findings into a conceptual scheme illustrating transcriptional regulation and signaling processes in leaf cells responding to long-term HL stress.

## 1. Introduction

Excess light constitutes a significant form of abiotic stress in plants. Even in regions with moderate illumination, fluctuations in light intensity can lead to reductions in crop yield. Increased light is detected by photoreceptors and triggers photoinhibition due to an imbalance between damage to the photosystem II and its repair, ultimately altering the cellular redox state [1]. Prolonged exposure to high-intensity light disrupts electron transport within the photosystems, resulting in the accumulation of reactive oxygen species (ROS) in chloroplasts [2]. This accumulation affects photosynthetic efficiency, suppresses growth processes, and triggers the biosynthesis of various secondary metabolites, including anthocyanins and phenylpropanoids [3], as well as the upregulation of antioxidant enzymes [4]. Moreover, it is now well established that ROS signaling intricately interacts with hormonal pathways involved in plant acclimation to adverse environmental conditions at the intracellular level. This interplay underlies widespread alterations in biomolecule concentrations at transcriptomic, proteomic, and metabolomic levels [5]. High light (HL) stress refers to the exposure of plants to excess light at intensities several times higher than normal (500–2000 µmol·$m^{-2}\cdot$$s^{-1}$), which induces chloroplast avoidance movement in *Arabidopsis thaliana* (L.) Heynh. [6] and occurs under natural sunlight conditions [7]. For instance, a three-day exposure to high light at 1200 µmol·$m^{-2}\cdot$$s^{-1}$ is non-lethal and allows full plant recovery, as previously demonstrated [8]. Most transcriptomic studies of HL stress in *A. thaliana* have been carried out within these intensity ranges, but with considerable variation in the duration of HL ranging from several minutes to several days.

The transcriptomic level of regulation is one of the fundamental layers in the implementation of genetic information. Quantitative analysis of the transcriptome and its comparison under different conditions is an effective approach to identify specific molecular-genetic systems and transcription factors (TFs) involved in plant stress responses. This strategy has also been applied to high light stress in *A. thaliana* [8]. To date, *A. thaliana* remains the primary model plant for which extensive data on transcriptomic responses to high light have been accumulated. In contrast, comparable data for other plant species remain limited, with only a few studies available, for example, in *Marchantia polymorpha* [9], *Chlamydomonas reinhardtii* [10], and *Citrus sinensis* [11].

In particular, a series of studies by Zandalinas et al. on the effects of short-term high-intensity light stress in *A. thaliana* [12,13,14,15], together with several investigations into transcriptomic responses to prolonged HL exposure (e.g., [16,17,18,19]), have significantly advanced our understanding of the responses of this plant to excess light under varying conditions of duration and intensity. These studies emphasized the importance of molecular interactions during the HL response of signaling pathways including hormonal, oxidative, and photoreceptor signaling, as well as the significant involvement of chloroplasts, spliceosomes, and other organelles. The large variety of these datasets could be used for the exploration of how experimental parameters (light intensity, time of treatment, etc.) influence transcriptomic outcomes. However, the broad diversity of available studies requires careful synthesis and systematic analysis for several reasons:The identification of differentially expressed genes (DEGs) is inherently probabilistic and may yield false positives, introducing uncertainty into the interpretation of results.The authors used diverse bioinformatics pipelines, software tools, and statistical thresholds for RNA-seq data processing, which complicates direct comparisons between studies.The individual studies employed different protocols for plant cultivation and stress exposure, and the biological materials analyzed (e.g., plant age, tissue type) also varied.

A standardized meta-analysis of stress-induced transcriptomes could help overcome the limitations mentioned above by allowing the identification of common patterns of transcriptomic responses. This type of analysis is most effective when it incorporates a broad and carefully curated selection of individual datasets. Previously, we demonstrated the utility of this bioinformatic approach applied to transcriptomic data on light stress responses in *A. thaliana*, incorporating a dataset comprising five independent experiments [20]. This analysis enabled the identification of several TFs (e.g., *PIF3*, *WRKY40*, *WOX11*, *HMGB10*) and biological processes (such as ribosome biogenesis, jasmonate signaling, flavonoid biosynthesis, and antioxidant gene activation) that are widely associated with excessive light exposure in photosynthetic tissues of *A. thaliana*.

Over the past three years since our previous analysis [20], numerous transcriptomic studies have been conducted to examine the long-term response of *A. thaliana* to high light stress. This has enabled a more refined classification and comparative analysis of DEGs involved in short- and long-term responses to excess light. In the current study, we improved our strategies for identifying relevant datasets and for bioinformatic analysis by processing all experiments uniformly from raw sequencing data. We had two main goals in our analysis: to compare the short- and long-term transcriptomic responses of plants to HL, and to identify key transcription factors in light-response gene regulatory networks. As a result, we collected a truly representative dataset of transcriptomic experiments that examine the effects of excess light on photosynthetic tissues of *A. thaliana*, allowing us to identify key genes and TFs involved in both immediate and sustained responses.

Our curated dataset encompasses 21 experiments covering 58 excess light conditions, representing 39 different combinations of time and intensity of HL. Altogether, we identified approximately 218,000 instances of DEGs, corresponding to around 19,000 unique *A. thaliana* genes. We introduce the publicly available, curated knowledge base AraLightDEGs (https://www.sysbio.ru/aralightdegs/, accessed on 16 July 2025, see Section 3.4), which enables users to efficiently explore light-responsive DEGs in *A. thaliana*. The database allows filtering by expression levels, frequency of DEG detection, gene identifiers, and Gene Ontology (GO) annotations, as well as by experimental parameters including light intensity, duration, tissue type, and plant age. Detailed metadata are provided for all included experiments, along with downloadable count matrices and DEG lists for individual conditions. In parallel, we present the AraLightMeta (https://github.com/av-bobrovskikh/AraLightMeta, accessed on 16 July 2025) pipeline implemented in R, designed for systematic meta-analysis, clustering, and gene regulatory network (GRN) reconstruction (see Section 3.5).

Using data from our knowledge base, we performed advanced bioinformatics analysis and reconstructed GRNs underlying both the early and long-term responses of *A. thaliana* to high light conditions. The initial stage of the main analysis (see Section 2.1) aims to classify the individual experiments and combine them into groups, as well as functional classification of the most frequent DEGs that are not specific to individual conditions. In the next step (Section 2.2), we revealed DEGs with robust regulation patterns in response to short- and long-term HL and examined in detail DEGs of the most relevant pathways according to the data from the literature. The final step of our analysis (see Section 2.3) aimed to find key TFs and their target genes by reconstructing their gene regulatory networks. This approach enabled us to identify key transcription factors involved in the plant’s long-term response to high light stress (see Section 2.4). In particular, we revealed novel gene interactions in response to high light, associated with the processes of chloroplast organization, the jasmonate and ROS pathways, and methylation.

## 2. Results and Discussion

### 2.1. The Initial Step of the AraLightMeta Analysis

#### 2.1.1. Classification of Experimental Conditions

To explore the transcriptomic responses of *A. thaliana* to high light (HL) conditions at the system level, we assembled and processed all available transcriptomic data. A total of 280 transcriptomic libraries, corresponding to 21 experiments, were included in the final dataset after initial selection and quality control. The raw count matrices for all experiments are accessible in the Downloads (https://www.sysbio.ru/aralightdegs/download, accessed on 16 July 2025) section of our developed resource and are also provided in Appendix A. To assess the primary effects of HL stress on plant transcriptomes, clustering was performed on 99 unique transcriptomic conditions, following the protocol described in Section 3.5.2. The resulting clustering is visualized as a dendrogram in Figure 1, while a version annotated with individual condition labels is presented in Appendix A.

Among the 99 conditions depicted in the dendrogram (Figure 1), there is an outlier cluster presented on the left, comprising two experiments, GSE251796 and PRJNA699408, totaling eight conditions. Due to pronounced batch effects, these two experiments were excluded from further analyses.

To the right of the outlier cluster (Figure 1), a separate cluster of medium-term HL transcriptomes (2 to 16 h) and long-term HL (1 to 3 days) is presented (13 conditions in total), marked in orange and red, respectively. Notably, this cluster does not include control samples, indicating pronounced transcriptomic alterations in response to medium- and long-duration HL treatments.

Mixed with the control samples, several experimental conditions with short HL exposures (marked in yellow) are present, along with three specific conditions: GSE137650 (500 µmol·$m^{-2}\cdot$$s^{-1}$ photons, for 6 and 30 h) and GSE236872 (500 µmol·$m^{-2}\cdot$$s^{-1}$ photons, for 4 h). This distribution suggests, on the one hand, that short-term HL treatments induce less pronounced shifts in plant transcriptomic programs than longer exposures. On the other hand, it also indicates that extended high light treatments at lower intensities may elicit a relatively modest transcriptional response.

In the dendrogram (Figure 1), samples derived from seedlings and leaves are distinguished by blue and green layers, respectively. These samples largely cluster in a tissue-specific manner; for example, the rightmost cluster exclusively comprises leaf samples. Overall, the short-response group (0–90 min) is enriched with diverse leaf samples, whereas the longer-response groups comprise approximately equal proportions of leaves and seedlings.

In general, the clustering results we obtained are consistent with previously described trends in the regulation of molecular mechanisms underlying plant responses to excess light. For example, prolonged high light exposure has been shown to induce a substantial accumulation of anthocyanins and flavonoids in the leaves of *Platycerium bifurcatum* [21], indicating a major metabolic reprogramming that is, at least in part, mediated by transcriptomic regulation. In a study included in our dataset (GSE111062), the authors reported fewer differentially expressed transcripts under short-term excess light stress compared to long-term stress [8]. Furthermore, previous studies have shown that early-responsive DEGs in plants are primarily associated with signaling pathways and hormone responses, whereas late-responsive DEGs are predominantly involved in metabolic processes [22]. Therefore, the primary factor influencing the clustering of samples is the duration of HL treatment. An overview of the experimental diversity of four main factors (duration of HL stress, its intensity, plant age, and tissue type) is presented in Figure 2.

In particular, in 18 of 33 experimental conditions for the short HL response (0–90 min), extremely high light intensities ranging from 1700 to 2000 µmol·$m^{-2}\cdot$$s^{-1}$ were applied (Figure 2A). Such intensities are not used in experiments with longer duration. The group of experiments under intensive HL conditions (1000–1330 µmol·$m^{-2}\cdot$$s^{-1}$) is relatively balanced in terms of the number of time points across all time groups (a total of 23 experimental conditions). In addition, there are 13 conditions of varying durations conducted under moderate HL intensity (500–900 µmol·$m^{-2}\cdot$$s^{-1}$, most of which pertain to responses lasting between 2 and 16 h. Overall, the dataset exhibits a marked quantitative bias toward short HL responses (33 out of 54 conditions). Of the 54 non-outlier experimental conditions, two-thirds are represented by leaf samples (Figure 2B). In general, the majority of the leaf experiments (27 out of 36 conditions) correspond to mature-stage leaves (21–30 days old). In contrast, only two conditions in seedlings correspond to the mature developmental stage.

To assess the similarity between leaves and seedlings in terms of transcriptomic profiles within the compiled dataset, correlations were calculated between the aggregated transcriptomic libraries derived from control samples, as well as from short HL (up to 90 min) and long HL (from 2 h to 3 days) treated samples from both tissues (Figure 2C). In general, seedlings and leaves exhibit a high degree of correlation; however, their dissimilarity is comparable to, and in some cases exceeds, the magnitude of changes induced by the stress response itself. For example, the correlation between control leaves and control seedlings is 0.94, while the correlation between control leaves and leaves treated with short HL reaches 0.97. These findings indicate a substantial tissue-specific effect, suggesting that the leaf and seedling subsets should be analyzed separately. We observed a relatively low correlation (0.88) between short- and long-term HL responses in plant leaves, which revealed the dynamic shift in the expression of multiple genes over the time of treatment. While the early HL response is mostly associated with ROS-associated signaling [12,14], prolonged HL treatment induced a diverse transcriptional response, including photosynthetic acclimation, modulation of hormonal and cytokinin signaling pathways, and suppression of growth-related processes [8]. This temporal divergence supports the general concept of plant stress response: an early response phase characterized by rapid redox signaling, followed by a system-level acclimation phase involving much more specific processes and TFs [23]. Furthermore, the higher transcriptional coherence observed in seedlings could be a sign of a less specialized stress response, consistent with their predominantly juvenile developmental stage and increased stress sensitivity. Given these distinct transcriptomic dynamics between short- and long-term responses in leaves, we focused our subsequent analysis on the leaves dataset, applying a clear temporal classification into short- and long-term HL response phases.

During the development of the knowledge base on the high light response DEGs AraLightDEGs (https://www.sysbio.ru/aralightdegs/, accessed on 16 July 2025), the described clustering of transcriptomic libraries was taken into account. The search functionality of the knowledge base includes two quick access buttons to identify DEGs associated with short-term responses (≤90 min; “Quick search for short response DEGs”) and medium-to-long-term responses (≥2 h; “Quick search for medium and long response DEGs”) to high light exposure. Furthermore, numerous advanced options have been implemented, including the ability to exclude experimental data from specific outlier experiments (PRJNA699408, GSE251796) via the toggle button “Remove outlier stress conditions from analysis”, as well as an option to restrict search results to TFs only (toggle button “Limit results to transcription factors only”).

The results obtained allowed us to determine the main strategy for the next steps:Focusing on the main analysis of leaf tissue. This dataset comprises twice as many experiments as those involving seedlings and is more balanced in terms of average plant age. In particular, 75% of the included experiments used plants in a mature developmental stage (21–30 days after germination, Figure 2B). This significantly reduces the batch effect associated with early developmental transitions.To perform the primary grouping of experimental conditions based on high light treatment duration, two groups were defined: short-term (up to 90 min) and long-term (2 h to 3 days). This classification reflects the observed transcriptomic patterns: responses to medium- and long-term HL (≥2 h) show high consistency during clustering, while short-term responses (≤90 min) exhibit heterogeneity and limited separation from control samples.Identification of consistent DEGs-specific regulations. Rather than relying on fold changes in individual experiments, our meta-analysis revealed DEGs with frequent upregulation and downregulation across multiple independent experiments.Exclusion of outlier datasets. The experiments PRJNA699408 and GSE251796 showed pronounced batch effects (Figure 1) and were removed from further analysis.

Therefore, our further analysis approach provides the robust protocol to reveal a core set of HL response DEGs in leaf tissue with focusing on direct comparison between short- and long-term responses DEGs and their corresponding gene regulatory networks.

#### 2.1.2. Functional Analysis of Most Frequent DEGs

Before analyzing short- and long-term specific DEGs, we explored the set of the most frequently differentially expressed genes in our database, defined as those detected in at least 50% of the experiments (corresponding to 27 experimental conditions). In total, we identified 978 DEGs. Among them, genes with a median log_2_fold change (log_2_FC) ≥0.5 were considered upregulated (498 genes), while those with a median log_2_FC ≤−0.5 were classified as downregulated (474 genes).

Within the group of upregulated DEGs (see Figure 3), there was a particularly strong and statistically significant enrichment for genes associated with ROS signaling (e.g., response to oxygen levels, response to oxidative stress), as well as genes involved in various types of stress responses, such as response to wounding, response to heat, and response to cold. Additionally, specific hormonal processes were enriched, including the response to jasmonic acid (JA) and the JA biosynthetic process. The regulation of the jasmonate pathway in response to HL stress was previously described in our earlier work [20]. Furthermore, the role of the salicylic acid pathway in light-responsive networks has been discussed in the study by [24]. The genes involved in the response to light intensity were also significantly enriched. The observed upregulation of processes such as protein maturation, folding, and refolding indicates a critical need to maintain proper protein folding under stress conditions, a requirement that has been emphasized in the review by [25]. Thus, the core set of upregulated genes in response to HL is largely associated with ROS-related processes, general stress-response pathways, and hormonal regulatory networks.

Among the downregulated genes, we observed those involved in the auxin pathway, cell communication, and signaling, which is primarily linked to the suppression of growth processes, potential disturbances in intercellular communication, and inhibition of photosynthesis. Evidence supporting the systemic regulation and interconnection of these processes has been presented in the work by [26]. Overall, such a transcriptomic response aligns with the phenotypic changes observed in *A. thaliana* subjected to HL stress, including leaf yellowing and growth inhibition [27].

### 2.2. The Second Step of the AraLightMeta Analysis

#### 2.2.1. Identification of Short-Term and Long-Term Specific DEGs

The HL-responsive leaf dataset we compiled enabled the classification of stress-induced transcriptomes into two groups, based on the obtained clusterization discussed in the previous section. Specifically, the short-term HL group includes 25 experimental conditions with light exposure ranging from 2 to 60 min, while the long-term HL group comprises 11 conditions with exposure durations from 2 to 30 h. To exclude unrobust DEGs, we selected only genes identified as differentially expressed in at least five leaf-specific experiments (see Figure 4A, dashed box). This filtering resulted in a set of 8510 DEGs for downstream analysis.

Subsequently, for each DEG, we calculated its frequency of occurrence in the short- and long-term HL groups, separately for upregulated and downregulated genes. Based on this, we defined time-specific DEG sets as those falling within the top quartile of frequency in each of the four categories: short-term upregulated, short-term downregulated, long-term upregulated, and long-term downregulated. Specifically, we found 2450 DEGs in the short-term upregulated group, 2954 DEGs in the short-term downregulated group, 2408 DEGs in the long-term upregulated group, and 2333 DEGs in the long-term downregulated group. In total, these sets encompassed 7001 unique genes, representing 82% of the total number of DEGs analyzed. Of these, 4234 DEGs (60%) were specific to particular time-regulation category, while 2398 DEGs (34%) were shared between two groups. The intersections between all groups of DEGs are illustrated in Figure 4B. Thus, our analytical approach demonstrated considerable specificity and informativeness, allowing us to identify nearly seven times more DEGs relevant to HL and refine their classification compared to our original methodology, which identified 1151 DEGs and provided a simplified classification into upregulated, downregulated, and mixed groups [20].

Next, to summarize the main biological processes relevant to the identified time-specific DEGs, we performed a GO enrichment analysis (Figure 4C). The largest number of enriched biological processes was observed for the upregulated DEGs under **short-term HL stress**. Notably, these processes are not strictly specific for HL response, but are associated with various stress response functions, ROS pathways, immune pathways, cellular signaling, and cell death. The accumulation of ROS under HL conditions is related to energy transfer through the electron transport chain of photosystem II in chloroplasts [28]. In severe short HL, it has been observed that there is a pronounced depletion of antioxidants and the accumulation of ROS, which triggers the transcriptional cascade of numerous stress-response genes, initiating the primary response [4]. Upregulation of pathways such as jasmonate signaling, rapid signaling pathways, and cell death TF *MYB30* during short-term HL stress was demonstrated in [12]. During short-term HL stress, a range of downregulated genes was observed, including those associated with leaf development, auxin response, cell wall organization, polysaccharide metabolism, and ribosomal biogenesis. Suppression of the auxin biosynthesis pathway under HL conditions was reported in [29]. Modifications of the cell wall have also been documented during long-term HL stress in *A. thaliana* seedlings [8], which may be partly explained by the tissue-specific nature of the temporal activation of this pathway.

**Long-term HL stress** is characterized by specific suppression of genes related to photosynthesis, development, the auxin signaling pathway (which is similar to short-term stress), and protein phosphorylation. The association of the long-term adaptive response of chloroplasts to HL with regulation of development, growth processes, and phosphorylation of light-harvesting complexes has been documented in previous studies [4,29], and our findings confirm this at the transcriptomic level. At the same time, our data indicate that under long-term HL stress, there is upregulation of pathways associated with ribosome biogenesis, amino acid biosynthesis, chloroplast organization, and RNA modifications. Adaptation of chloroplasts to high light involves a multistep recovery process of photosystem II, accompanied by subcompartmentalization of thylakoids into stacked grana [30]. Previous research has shown that ribosome biogenesis is suppressed under prolonged high light exposure (450 µmol·$m^{-2}\cdot$$s^{-1}$) in *A. thaliana* [31], while our results demonstrate a stable upregulation of this process during the long-term response and its suppression in the short-term response. Differences in observed regulatory patterns may have arisen from variations in the experimental design and the use of insufficiently intense HL treatment in the cited study. However, our data reflect the tendency that this process is specifically regulated under HL conditions and, in our large dataset of long-term HL experiments, it is indeed upregulated. N6-methyladenosine RNA modifications are known to play a role in the regulation of photosynthesis under HL stress in *A. thaliana* [32]. Furthermore, amino acid biosynthesis during pretreatment irradiation has been reported to help *A. thaliana* plants cope more effectively with cadmium stress, indicating the adaptive role of this mechanism under HL stress as well [33].

Taken together, our findings align with prior research and refine the temporal dynamics of pathway regulation under high light stress. It should be noted that many acute stress response and signaling pathways upregulated during short-term HL stress are, in contrast, downregulated during the long-term HL response. This observation suggests a pronounced transcriptomic switch in plant adaptation over time, transitioning from short-term, non-specific ROS- and stress-associated responses to long-term acclimation mechanisms.

#### 2.2.2. Regulation of Key Pathways Related to Long-Term High Light Response

According to current understanding of the systemic nature of the HL response, oxidative stress in chloroplasts triggers retrograde signaling pathways that modulate expression of nuclear genes [34]. Photoreceptors acts in a coordinated manner with retrograde signals to coordinate HL response, leading to suppression of photosynthesis-associated nuclear genes (PhANGs) [34]. In addition, key regulatory genes such as *GUN1*, *BBX16*, and *GLK1* are noted to play important roles in this process [34]. Furthermore, the ROS accumulation in chloroplasts induces jasmonate-related retrograde signaling, connecting crosstalk between redox and hormone pathways [35]. Ultimately, light-induced transcription networks drive changes in protein metabolism and photomorphogenenic development [36]. Taking this into account, we aimed to identify the expression patterns of key HL-responsive DEGs across these categories. We focused on revealing DEGs with consistent upregulation or downregulation in at least 50% of the experimental conditions under short- or long-term HL treatments. For TFs, we applied a stricter threshold of 80% due to their large numbers, aiming to select only the most relevant TFs. The heatmaps showing robust DEGs among these categories are presented in Figure 5. Notably, consistent regulation of such DEGs is predominantly observed under long-term HL conditions. Below, we discuss the main findings for these genes under prolonged stress conditions.

**Antioxidant genes.** Serving as key regulators of local ROS levels in plant cells [37], antioxidant genes show divergent regulation under long-term HL stress, with approximately half being upregulated. Among the **upregulated** genes are those encoding the iron superoxide dismutases Fe-SOD (1–3), dehydroascorbate reductases *DHAR1* and *DHAR2*, several glutathione S-transferases (GSTs) (*GSTU5*, *GSTU20*, *GSTF6*, *GSTF12*, *AT5G42150*), stromal ascorbate peroxidase (*SAPX*), cytosolic ascorbate peroxidase (*APX1*), and chloroplast-localized glutathione peroxidase *GPX7*. It has been shown that *DHAR1* and *DHAR2* participate in modulating the redox states of the ascorbate–glutathione cycle in response to HL stress [38]. Previous studies established that superoxide dismutases are activated under HL conditions; our results further specify that this trend is particularly evident for iron-dependent SODs [39]. Conversely, several antioxidant genes are **downregulated** during HL stress, including thylakoid ascorbate peroxidase (*TAPX*), chloroplast glutathione peroxidase *GPX1*, catalase *CAT3*, and several GSTs (*GSTU18*, *GSTU27*, *GSTF7*, *GSTF8*). It has been demonstrated that knockout *A. thaliana* plants lacking stromal and especially thylakoid ascorbate peroxidases exhibit pronounced stress phenotypes, accumulating hydrogen peroxide and oxidized proteins [40]. This observation is particularly intriguing in the context that thylakoid ascorbate peroxidase is downregulated under HL stress, which may suggest a unique role for thylakoid-localized retrograde signaling in regulating stress-associated nuclear gene expression. The involvement of several GSTs (*GSTU5*, *GSTF2*, *GSTU17*, *GSTU20*, *GSTU26*, *GSTU28*) in light-dependent signaling has been highlighted in the review by Gallé et al. [41]. Our data indicate an even broader role for Phi-class GSTs and their potential involvement in HL-induced redox signaling.

**Redoxins.** The role of chloroplast thioredoxins in redox signaling pathways under fluctuating light conditions is well established [42]. In our analysis, we identified consistent upregulation of several chloroplast thioredoxins during long-term HL stress, including *TRX z*, *AT2G31840* (thioredoxin-like), and *AT2G37240* (thioredoxin-like), as well as two mitochondrial thioredoxins (*AT5G57230* and *TO1*). Additionally, other chloroplast thioredoxins such as *ATHM3*, *TRXF2*, *ACHT4*, *AT1G07700*, and two membrane-bound TPR repeat-containing thioredoxins (*TTL3* and *TTL4*) were upregulated. The photoprotective function of glutaredoxin *GRXS13* in *A. thaliana* has been previously described [43]. Under long-term HL conditions, our data show downregulation of five cytosolic glutaredoxin genes: *GRXS2*, *GRXS10*, *GRXC1*, *GRXC11*, and *AT5G13810*. We also observed upregulation of two chloroplast ferredoxins (*FD1*, *FD3*) and mitochondrial ferredoxins *MFDX2* and *AT3G27570* (Sucrase/ferredoxin-like family protein). It has been shown that ferredoxin:NADP(H) reductase is involved in the photoprotective functions of photosystem I under light stress [44]. Therefore, the upregulation of ferredoxins detected in our analysis might reflect a complementary functional requirement. The functions of cupredoxins and peroxiredoxins under HL stress in plants remain largely unexplored in the literature. However, we found consistent upregulation of two peroxiredoxins (*AT3G52960* and *TPX1*) and the cupredoxin *LPR1*, alongside downregulation of two cupredoxins (*AT4G12420* and *AT1G20340*).

Taken together, the transcriptional dynamics of redoxins and antioxidant genes suggest a complex response and compartment-specific interplay of ROS signaling across cellular organelles [37]. Moreover, these findings indicate differential functional importance among antioxidant and redox enzymes, as well as substantial variation in regulation between various genes within the same class, as previously demonstrated [45].

**Photosynthesis-associated nuclear genes.** The vast majority of genes encoding components of the light-harvesting complex and photosystem II subunits are downregulated under long-term HL conditions, consistent with previous findings [34]. However, it is interesting that the expression of three genes (*RBCS1B*, *LHCB4.3*, and *PSB28*) is **upregulated** in response to light stress. Protein abundance of *LHCB4.3* increases within one hour after high light exposure, suggesting a possible photoprotective function [46]. In the case of *PSB28*, it has been demonstrated that this protein is required for the stable accumulation of core complexes of photosystem II in *A. thaliana*, which may explain the importance of its transcriptional upregulation during HL stress [47].

**B-box proteins** are important regulators of light-dependent plant development, functioning as TFs that not only regulate target genes but also interact with a range of other factors involved in the regulation of plant development [48]. According to our data, the genes *BBX8*, *BBX13*, *BBX18*, *BBX29*, *BBX31*, and *BBX32* are upregulated under prolonged HL exposure, whereas *BBX14*, *BBX15*, *BBX17*, *BBX19*, *BBX21*, and *BBX27* are downregulated. Among the **upregulated** B-box genes, several light-specific functions have been documented in *A. thaliana*: *BBX8* promotes shade avoidance by activating *PIF4* [49]; *BBX13* suppresses photoperiodic flowering [50]; *BBX18* acts as a regulator of thermomorphogenesis, contributing to plant adaptation to elevated temperatures often accompanying excessive light [51]; *BBX29* and *BBX31* together with HY5 form a feedback loop to fine-tune photomorphogenic development [52]; and *BBX32* is implicated in the regulation of acclimation to HL in mature leaves [53]. Among the **downregulated** genes, *BBX14* is known to regulate chlorophyll biosynthesis during early light exposure and functions as a circadian clock component, being target of *GLK1* [54]. The roles of *BBX15* and *BBX17* in light-response processes remain insufficiently characterized, although they also exhibited downregulation in response to HL in the study by Huang et al. [8]. *BBX19* promotes degradation of ELF3 through its interaction with *COP1*, which is confirmed by our findings as both *ELF3* and *ELF4* are upregulated [51]. *BBX21*, together with *BBX11* and *HY5*, is involved in a positive feedback loop of photomorphogenic development [55]. Finally, *BBX27* appears to be a potential downstream target of *BBX31* and *HY5*, with its expression levels decreasing after dawn [56].

**Photoreceptors.** Most photoreceptor genes, as well as two main phytochrome-interacting factors (*PIF4* and *PIF7*), were found to be downregulated under long-term HL conditions, which is consistent with the findings of Huang et al. [8]. However, six photoreceptor-related genes were **upregulated**: *ELF3*, *ELF4*, *TOC1*, *PRR5*, *HY5*, and *HYH*. It is known that physical interactions among evening clock components, such as *PRR5*, *TOC1*, and *ELF3*, establish a repressive chromatin structure at the *PIF4* locus, thereby inhibiting hypocotyl elongation during the night [57]. The *ELF4* protein may also possess complementary functions in this regulatory network. Additionally, the TFs elongated hypocotyl *HY5* and *HYH* are involved in the long-term maintenance of the shade avoidance response and suppress the expression of *XTH* genes associated with cell wall loosening. Thus, their upregulation under HL conditions contributes to inhibition of hypocotyl growth [58].

**Photomorphogenesis.** This category of genes partially overlaps with the photoreceptors and B-box proteins discussed above. Six genes were **upregulated** during the long-term HL response: *UAP56b*, *CSN6A*, *AT4G34550*, *AT5G06550*, *LZF1*, and *ATAF2*. Moreover, *LZF1* and *ATAF2* are more pronouncedly upregulated in the short-term HL response. It is known that the RNA helicase *UAP56* interacts with *COP1* to regulate alternative splicing, thus repressing photomorphogenesis [59]. *CSN6A* is a subunit of the COP9 signalosome, which is required for the nucleocytoplasmic distribution of *COP1* within the light signaling pathway [60]. *AT4G34550 (PCHL)* is involved in the control of photomorphogenesis by suppressing the dark reversion of phytochrome B and enhancing light sensitivity in plants [61]. *AT5G06550 (JMJ22)* is activated by red light and participates in the induction of gibberellic acid biosynthesis genes [62]. *LZF1* has a well-established role in light-dependent development and is regulated by *COP1* [63]. *ATAF2* is a transcription factor that modulates photomorphogenesis in seedlings, interacting with phytochrome A and the circadian regulator *CCA1* [64]. In contrast, nine genes involved in photomorphogenesis were found to be **downregulated**: *ABCB19*, *AMP1*, *AT5G63080*, *CKL4*, *NF-YC1*, *NF-YC4*, *PIL5*, *PRE1*, and *SBH2*. *ABCB19*, an auxin transporter, influences light perception in plants [65]. *AMP1* impacts guard cell aperture and photosynthetic efficiency [66]. *AT5G63080* (JMJ20) shares functional similarity with *JMJ22* [62]; however, their opposing regulation under HL conditions warrants further investigation. The *CKL4* gene phosphorylates two photoreceptors, *TOC1* and *PRR5*, enhancing their ability to repress *PIF4* [67]. NF-YC proteins are major regulators of light-dependent growth and developmental processes in plants, including suppression of the brassinosteroid pathway during hypocotyl elongation in light conditions [68]. *PIL5* (also known as *PIF1*) is involved in the phytochrome light-dependent signaling pathway [69]. *PRE1* is a target gene of the transcription factor *PIF3*, which plays a role in promoting hypocotyl elongation [70]. Finally, *SBH2* is a component of the sphingolipid biosynthesis pathway, although its specific role in the HL response in *A. thaliana* remains unknown.

**Chloroplast organization**-related DEGs are **consistently upregulated** under long-term HL conditions, representing one of the key intracellular adaptations of plants to excessive light exposure [30]. This process is among several that exhibit specific upregulation during long-term HL stress, as confirmed by the functional analysis discussed in the previous section of the results. Nearly 50 such genes were identified, exhibiting coordinated transcriptional upregulation during the long-term HL response.

**Transcription factors**. We identified several TFs that appear to play key roles in the long-term HL response. Given that prolonged HL stress leads to suppression of photosynthesis, developmental processes, and photomorphogenesis (as discussed in a previous section), we observe a greater number of downregulated TFs in our list. However, those TFs that are **upregulated** are likely to be most important regulators of stress acclimation and survival: *GBF6*, *HD2C*, *HDA3*, *PAP1*, *SPL11*, *TRFL3*, *ATWHY2*, *WHY3*, *AT4G12750*, and *AT4G16750*. Recent work discusses the role of *GBF6* (also known as *bZIP11*) as a potential repressor of genes encoding the photosynthetic light-harvesting complex and as an inducer of polysaccharide catabolism and starvation responses, which is consistent with our findings [71]. It is known that the histone deacetylase *HD2C* physically interacts with the RPD3-type histone deacetylase *HDA6* and is involved in abscisic acid and stress signaling pathways; however, its function under high light remains uncharacterized [72]. It is plausible that similar interactions might exist between *HD2C* and *HDA3*. The phytochrome-associated TF *PAP1* plays a key role in the regulation of the anthocyanin biosynthesis pathway and is repressed by the TF *PIF4* [73]. Although there is no specific information about the role of *SPL11* in HL stress, its paralogs are known to contribute to plant thermotolerance [74]. *TRFL3* is referenced in the context of a functional association between the LDL1/2-HDA6 modification complex and *CCA1*/*LHY* in the regulation of circadian clock genes, where it acts as a target gene of *CCA1*. The plastid protein *WHIRLY1* is known to participate in light adaptation and is associated with ROS signaling networks [75]. In our data, we detected consistent upregulation of its paralogs, *ATWHY2* and *WHY3*, suggesting that all of these genes may play a significant role in fine-tuning ROS-associated networks. The homeodomain-like TF *AT4G12750* has been identified as a core component of the retrograde gene expression network in chloroplasts and is strongly upregulated under HL conditions [76]. Finally, the ERF-family TF *AT4G16750* has been predicted as one of the top 20 important regulators involved in responses to multiple stresses [77].

Thus, all major pathways described in the literature exhibit pronounced transcriptional dynamics in response to high light, and our extensive transcriptomic dataset allowed us to characterize consistent regulatory patterns for individual genes. On the other hand, several less obvious but potentially important regulators of the stress response can be identified using approaches to reconstruct gene regulatory networks [20,78]. Given the extensive dataset compiled in AraLightDEGs, we focused the final stage of our analysis on inferring potential TF–target gene interactions using coexpression analysis and regulatory network inference approaches.

### 2.3. The Third Step of the AraLightMeta Analysis

#### 2.3.1. Gene Regulatory Network of Long-Term High Light Response

In reconstructing GRNs, we used a combined approach integrating weighted gene coexpression network analysis (WGCNA) across all genes and GENIE3-based regulatory inference between TFs and non-TF genes. The final networks include only top-ranking interactions supported by both methods and meet empirical thresholds: a coexpression of ≥0.3 for the long-term HL GRN and ≥ 0.5 for the short-term HL GRN, and permutation *p*-value <0.05 for regulatory inference.

For the long-term HL response dataset (31 transcriptomic libraries in total, 2–30 h of treatment), we identified a GRN consisting of 11 TFs connected to 114 target genes via 162 edges (0.3–0.47 coexpression correlations). This network forms four distinct clusters (Figure 6).

The **first cluster**, consisting of a total of 44 genes, is associated with jasmonate signaling, genes involved in responses to fatty acids, and several antioxidant genes. The importance of the jasmonate pathway in acclimation to the combined stresses of high light and heat in *A. thaliana* has been highlighted in the study [16]. Specifically, this cluster includes eight genes from the jasmonate pathway: *LOX3*, *JAZ5*, *JAZ1*, *BT4*, *AT5G05600*, *AT3G51450*, *AT2G34810*, and *ADC2*, as well as ten stress-response genes: *TXR1*, *RD26*, *PTF1*, *ILL5*, *GSTU5*, *ERD9* (also known as *GSTU17*), *CP1*, *CLH1*, *AT5G53750*, and *AT4G35110*. Four TFs are present in this cluster: *CRF3* (upregulated), *PTF1* (downregulated), *GATA17* (upregulated), and *RD26* (downregulated), with the first two TFs linked to the largest number of potential target genes. Notably, our analysis identifies, for the first time, TFs *CRF3* and *GATA17*, that may regulate this set of genes, which are not documented in major databases such as STRING (https://string-db.org, accessed on 16 July 2025). Additionally, this cluster contains previously unknown strong negative correlations between the downregulated TFs *PTF1* and textitRD26 and their upregulated target genes, suggesting that both TFs may act as repressors of JA- and stress-responsive pathways genes in normal conditions.

The **second cluster**, with 36 upregulated genes in total, is significantly associated with ribosome biogenesis and RNA processing, processes that are specific to the long-term HL response (see Figure 4). Notably, 18 genes are associated with both terms: *NOF1*, *La1*, *EMB2777*, *AT5G54910*, *AT5G35910*, *AT5G15550*, *AT5G14520*, *AT5G11240*, *AT4G04940*, *AT3G57940*, *AT2G40360*, *AT2G34357*, *AT1G77030*, *AT1G69070*, *AT1G42440*, *AT1G23280*, *AT1G10490*, and *AT1G06720*. Additionally, two genes, *NRPA1* and *AT5G17930*, are specifically linked to ribosome biogenesis, while three genes, *ELP2*, *AT5G39840*, and *AT2G28450*, are associated with RNA processing. This cluster includes three TFs: *AT2G28450*, *AT4G00238*, and *AT4G39160*. *AT2G28450*, encoding a zinc finger protein, exhibits the highest connectivity and, therefore, can be considered as the central hub of this cluster. In our previous work, we also revealed DEGs associated with ribosome biogenesis in response to HL, although we did not explicitly elucidate their regulatory network [20]. While protein–protein associations among most genes in this cluster are documented in the STRING database, our GRN analysis predicts novel regulatory interactions between the TF *AT4G00238* and five genes involved in ribosome biogenesis and RNA processing (*La1*, *AT5G15550*, *AT5G14520*, *AT1G23280*, *AT5G11240*), as well as with *RIN1*. Therefore, our findings extend the current understanding of the regulatory landscape underlying ribosome biogenesis and RNA metabolism under HL stress.

The **third cluster**, comprising 31 upregulated genes, is predominantly composed of genes that encode proteins located in the chloroplast. This group includes 17 chloroplast-targeted genes: *PTAC17*, *PTAC14*, *PTAC12*, *PDE312*, *MRL7-L*, *FLN2*, *EMB3120*, *EMB3108*, *emb2746*, *EMB2219*, *CPZ*, *Cpn60alpha2*, *AT5G66470*, *AT3G23940*, *AT3G18680*, *AT2G44640*, and *AT2G19870*. Additionally, the cluster contains seven genes with dual chloroplast-mitochondrial localization, namely, *WHY3*, *RH39*, *NOA1*, *HSP60-2*, *GYRA*, *BSM*, and *AT5G53920*, as well as two mitochondrial genes, *ATWHY2* and *pde194*. This cluster includes four TFs: *WHY3*, *ATWHY2*, *emb2746*, and *OCP3*, with the first three acting as central hubs in the network. According to the STRING database, functional protein–protein associations exist between *ATWHY2* and *WHY3* and several identified target genes, supporting their role in chloroplast organization. Our data suggest that the upregulation of these TFs under long-term HL stress may facilitate retrograde signaling and chloroplast–mitochondrial crosstalk, contributing to the acclimation of chloroplast function under excessive light. Importantly, we report, for the first time the potential existence of shared targets (namely, *AT2G19870* and *AT3G23940*) between *WHY3* and the TF *emb2746*. The gene *emb2746* (also known as *RNJ*) has been previously described as essential for chloroplast and embryo development in *A. thaliana* [79]. Furthermore, it has been shown that the TF *VIR*, required for the accumulation of photoprotective proteins under HL conditions, physically interacts with *RNJ* to support RNA processing and stability [32]. Therefore, the interactions between chloroplast-dependent genes and these three TFs represent a biologically coherent finding in our study.

The **fourth cluster**, comprising 14 upregulated genes, is associated with processes of RNA methylation. It includes four methyltransferases: *AT4G04670* (tRNA wybutosine-synthesizing protein 2 homolog), *AT4G26600* (26S rRNA (cytosine-C(5))-methyltransferase NOP2B), *AT4G27340* (tRNA (guanine(37)-N1)-methyltransferase 2), and *AT4G40000* (S-adenosyl-L-methionine-dependent methyltransferases superfamily protein). Additionally, the cluster contains four genes related to aromatic compound biosynthesis: *AT1G63660*, *AT1G79470* (IMPDH), *AT4G12750* (RLT3), and *NRPC2*. This cluster includes a single TF, *RLT3*. Notably, no known functional associations between RLT3 and the other genes in this cluster were found in the STRING database. Thus, our analysis reveals a potential interplay between RNA methylation and the biosynthesis of aromatic secondary metabolites during long-term HL acclimation. Recent studies have emphasized light-dependent regulation of the methylome in *A. thaliana* [70,80]. Furthermore, wybutosine, a modified base in tRNA, is recognized as a stress-responsive guanine derivative in eukaryotes, implicated in maintaining translational fidelity under adverse conditions [81], suggesting that tRNA methylation may play a critical role in stress resilience in plants. As discussed earlier, *RLT3* is part of the core retrograde signaling network coordinating chloroplast gene expression under high light stress [76].

In summary, the reconstructed GRN underlying the long-term HL response explicitly reveals the key transcriptional programs associated with organelle acclimation and adaptation to systemic stress. The first cluster is associated with the upregulated JA signaling pathway, which regulates photomorphogenetic processes by suppressing hypocotyl elongation [82]. Notably, this cluster suggests a potential interplay between the JA and ROS signaling pathways, as TFs (*CRF3*, *PTF1*) are predicted to co-regulate both JA-responsive genes and two GSTs. The light- and JA-dependent modulation of *GSTU17* expression and its role in *A. thaliana* seedling development have been reported previously [83]. Our findings extend this knowledge by identifying specific TFs that may coordinate expression of JA pathway and both GST genes. The second cluster, associated with ribosome biogenesis and RNA processing, reflects a long-term reorganization of the translational machinery. This indicates a shift in cellular resource allocation toward the synthesis of stress-adaptive proteins, including those involved in amino acid biosynthesis, chloroplast function, photoprotection, and redox homeostasis. The third cluster shows coordinated upregulation of genes localized to both chloroplasts and mitochondria, including regulatory hubs *WHY3* and *ATWHY2*, underscoring the importance of inter-organellar communication during acclimation. Finally, the fourth cluster includes methyltransferases involved in RNA modification and genes related to aromatic compound biosynthesis, with RLT3 (AT4G12750) acting as the single TF, which was not previously known.

#### 2.3.2. Gene Regulatory Network of Short-Term High Light Response

For the short-term HL response dataset (75 transcriptomic libraries in total, 2–60 min of treatment), we reconstructed a GRN which consist of 16 TFs connected to 50 target genes via 64 edges (0.5–0.65 coexpression correlations). This network forms three distinct clusters (Figure 7). Importantly, the majority of genes (58) within this network are downregulated, suggesting a predominant repressive transcriptional program during the early phase of HL stress.

The **first cluster** consists of 38 genes, including 10 TFs, of which only 2 are upregulated (*MYB6* and *GBF2*). The highest network is observed for TFs *REV*, *KNAT7*, *SHR*, and *MYB6*, suggesting their hub roles. Functionally, this cluster is enriched in growth-related processes, hormone signaling pathways, and macromolecule biosynthesis. It includes *SHR*, which is involved in radial pattern formation; the metabolic regulator gene *SPL10*; and *REV*, which participates in both processes. Additionally, several genes are associated with macromolecule biosynthetic pathways, including *UDG4*, *MYB6*, *MYB4*, *MSR1*, *KNAT7*, *GBF2*, *F8H*, *ARF8*, *AL5*, *AT5G14370*, *AT4G29100*, and *AT3G06590*. The specific roles of these genes in the HL response remain uncharacterized. We hypothesize that during the early phase of HL stress, plants do not activate targeted transcriptional programs but, instead, undergo a broad downregulation of growth- and metabolism-associated genes, consistent with a resource reallocation strategy toward stress survival [23]. This is reflected in the GRN structure, where top-ranked interactions predominantly involve downregulated metabolic and developmental regulators.

The **second cluster** comprises 16 genes and shows no significant GO term enrichment. It includes 3 downregulated TFs (*ATHB-15*, *AT3G61950* (bHLH67), and *AT5G05790*) and 13 of their potential target genes.

The **third cluster** consist of 12 genes and features a single *AT1G69580* (*PHL8*), along with 2 genes involved in cell wall growth and multidimensional cell expansion: *PG2* and *GH9B1*.

Thus, in the reconstructed GRN consisting of the top-ranking interactions under short-term HL conditions, no significant overlaps were identified with short-specific processes (see Figure 4), such as cell death, response to jasmonic or salicylic acid, or protein folding. This contrasts with the pronounced transcriptional reprogramming observed during long-term HL, as discussed in the previous section. Moreover, the absence of well-established HL stress pathways in the short-term GRN aligns with previous studies indicating that brief HL exposure primarily activates ROS-associated signaling rather than full-scale acclimation responses [12,14]. This supports the interpretation that early transcriptional changes are driven by redox signals rather than direct photodamage or long-term photoprotective mechanisms [84]. In this context, preferential downregulation of genes involved in histogenesis (first cluster) and cell wall modification (third cluster) may reflect ROS-mediated suppression of growth-related processes, which is a well-documented trade-off during early stress responses [84]. Such repression is thought to redirect cellular resources toward survival and maintenance, even before significant structural damage occurs.

Nevertheless, our data reveal a small subset of genes from key adaptive systems that are clearly upregulated under short-term HL but not under long-term HL. These include the thioredoxin *TRX5*, antioxidant genes such as *GSTF8*, *GSTF7*, and *CAT3* (which, conversely, are downregulated under long-term HL), the circadian clock regulator *TOC1*, two photomorphogenesis-related genes *LZF1* and *ATAF2*, and the TF *ZAT6* (see Figure 5). However, these genes do not show strong coexpression with growth-related modules in the short-term HL GRN. At the same time, the antioxidant genes listed above are linked to ROS-associated signaling pathways. This observation indicates that the short-term HL transcriptional response is relatively nonspecific in its regulatory architecture and primarily oriented toward a rapid oxidative burst, rather than sustained acclimation. This supports the idea that even in the absence of strong pathway-level enrichment, the short-term response reflects a biologically meaningful redox-driven regulatory program [12,14]. Therefore, although the short-term HL response lacks the diverse, multi-component regulatory architecture observed in the long-term GRN, it may serve as a valuable transcriptional model for identifying candidate regulatory interactions within ROS-associated signaling networks.

### 2.4. Conceptual Scheme of the Transcriptional Response of *A. thaliana* Leaves to Long-Term High Light

Given the identification of several biologically meaningful regulatory interactions associated with the long-term response to HL, we aimed to reconsider existing models of systemic regulation in photosynthetic cells under HL conditions. As a conceptual foundation, we refer to three schematic models presented in [34], which summarize the central role of chloroplast-derived retrograde signaling in mediating organelle-to-nucleus communication under HL and other stresses. These models emphasize the suppression of photosynthesis-associated nuclear genes, and highlight key components such as photoreceptors, ROS, and several crucial regulatory proteins including *GUN1*, *GLK1*, *BBX16*, *CRY1*, and *HY5* [34]. Based on our findings in Section 2.3.1 and these models [34], we constructed an integrated conceptual scheme that depicts regulatory pathways and gene interactions in *A. thaliana* photosynthetic cells in response to long-term HL stress (Figure 8).


**Key findings:**
**Shared regulation of glutathione S-transferases and jasmonic acid pathway genes:** We predict, for the first time, that two GSTs (*GSTU5*, *GSTU17*) are co-regulated with seven JA-response genes (*JAZ1*, *JAZ5*, *LOX3*, and others) via the upregulated TF *CRF3* and the downregulated *PTF1*, which could act antagonistically (GRN Cluster 1, Figure 6). This raises **Q1**: whether other upregulated GSTs indirectly associate with the JA pathway.**Ribosome biogenesis regulation:** TF *AT2G28450* is predicted to regulate ribosome biogenesis genes, which are consistently upregulated during long-term HL stress (GRN Cluster 2, Figure 6).**Chloroplast–mitochondrial interplay:** We revealed HL-induced upregulation of organelle-specific genes, mediated by TFs *ATWHY2*, *WHY3*, and *emb2746*, with the latter being shown for the first time to have this functional role (GRN Cluster 3, Figure 6).**Methyltransferase upregulation:** *AT4G04670*, *AT4G27340*, *AT4G40000* are consistently upregulated, and potentially controlled by TF *AT4G12750* (GRN Cluster 4, Figure 6).


While Griffin and Toledo-Ortiz in 2022 [34] highlighted the proteins GUN1 and GLK1 in chloroplast retrograde signaling, we observed no significant regulation of these genes under HL. Instead, their paralogs, *GUN5* and *GLK2*, were downregulated. Given the proposed inhibitory role of the GUN1–GLK1 complex on *BBX16*, which causes inhibition of photosynthesis-related genes [34], we pose **Q2**: whether *GLK2* and *GUN5* influence regulation of certain B-box genes. Notably, six B-box genes were upregulated and six downregulated, with established links to circadian regulators *HY5* and *ELF3* (highlighted in Figure 8) [49,51,52,55,56]. Additionally, most circadian rhythm genes were upregulated, except *LHY*.

Consistent with previous works, photoreceptor and photosynthesis-associated nuclear genes were suppressed [8,34]. However, three photosynthesis-related genes (*RBCS1B*, *LHCB4.3*, *PSB28*) were upregulated under HL stress, whereas the majority are downregulated; this is posed as **Q3**.

The presented scheme has certain limitations and does not capture the full complexity of multi-level interplay during the HL response. Our model is based on identified DEGs and GRN-derived interactions, which inherently reflect gene expression dynamics at the leaf tissue level but do not directly account for pH-dependent chloroplast signaling during acidification of the thylakoid lumen and stromal alkalization [85], rapid post-translational modifications [86], and photosystem II repair processes that shape the early stage of light-dependent response [87]. The conceptual representation of our scheme as “photosynthetic cell” was introduced to integrate nuclear and chloroplast responses within a unified model, and it could be further clarified by using single-cell RNA sequencing data directly for photosynthetic cells [88]. Furthermore, we acknowledge that gene expression and signaling under HL stress are governed by a complex interplay of physical factors (various spectrum and wavelength of light used [89]), chemical factors (carbohydrates, phenolic, chloroplast content, and other substances [90]), and spatial factors (signal transduction processes occurring between cell layers and light-dependent leaf movement [91]) that are not explicitly represented in our model. Nevertheless, our work identifies novel transcriptional regulators and their interactions (highlighted in green in Figure 8), elucidates key regulators within known networks, and clarifies distinct gene regulation patterns in response to HL stress.

### 2.5. Main Findings of Transcriptomic Response of Seedlings to High Light

Given that leaves are the primary photosynthetic organs and the majority of our data are derived from leaf datasets, we focused our investigation on leaf responses to high light stress. However, we extended our analysis to seedlings to assess the conservation of stress responses across photosynthetic tissues. We obtained partially similar results for certain genes and biological processes involved in HL response to seedlings.

A Sankey diagram of top GO-enriched terms in seedlings is provided in Appendix A. Notably, seedlings exhibited greater coherence between short- and long-term HL responses in terms of shared biological processes. As in leaves, long-term upregulated DEGs in seedlings were enriched for plastid organization, ribosome biogenesis, and rRNA processing, while short-term upregulated DEGs were associated with JA and fatty acid responses.

Conversely, downregulated processes common to both short- and long-term HL in seedlings included response to auxin (consistent with leaf responses), cell wall organization (specific to short-term responses in leaves), protein phosphorylation (associated with long-term responses in leaves), and immune response (upregulated short-term but downregulated long-term in leaves). When compared to leaves, it is evident that response to auxin is similarly represented in both the short- and long-term HL responses.

Overall, fewer top enriched terms were identified in seedlings compared to leaves (38 versus 47), and, notably, there was no significant downregulation of photosynthesis-related terms specifically enriched under long-term HL conditions in seedlings. Heatmaps illustrating the expression profiles of key pathways in seedlings are provided in Appendix A. As with leaves, a greater number of relevant DEGs were identified in the long-term response than in the short-term response.

Reconstructed gene regulatory networks from seedling data (Appendix A) revealed higher correlated gene expression than in leaves, suggesting a more systemic but less specific stress response. Taken together, we conclude that the seedling dataset largely confirms the major trends observed in the transcriptomic response of leaves; however, it is less detailed due to a higher level of noise arising from tissue and cell-type heterogeneity inherent in seedlings. Despite these limitations, the seedling dataset remains valuable for cross-stress comparative studies in *A. thaliana* exposed to different types of stresses. However, it is important to note that the seedling dataset obtained is not well balanced in terms of plant age, as it predominantly includes very young and older seedlings (see Figure 2B), a factor that should be carefully considered in future comparative studies.

## 3. Materials and Methods

### 3.1. Search and Selection of Relevant Transcriptomic Experiments for Analysis

A systematic search and selection of publicly available transcriptomic datasets for meta-analysis was conducted using the NCBI GEO repository (https://www.ncbi.nlm.nih.gov/geo/, accessed on 19 January 2025). The datasets were required to meet the following four criteria:The organism studied was *A. thaliana*, genotype Columbia (Col-0);The experiments analyzed transcriptomes of photosynthetic tissues;The stress condition involved exposure to excess light, with a minimum light intensity threshold of 500 µmol·$m^{-2}\cdot$$s^{-1}$ up to 2000 µmol·$m^{-2}\cdot$$s^{-1}$ and up to five days of exposure duration;Each dataset included control groups of plants grown under normal light conditions, defined as light intensities ranging from 50 to 150 µmol $m^{-2}$ $s^{-1}$.

The search was carried out using the keyword “high light”, with filters applied for the *A. thaliana* (taxon ID: 3702) and the study type set to “Expression profiling by high-throughput sequencing”. To ensure completeness of dataset selection, additional experiments were screened in the SRA (https://www.ncbi.nlm.nih.gov/sra, accessed on 19 January 2025), BioProject (https://www.ncbi.nlm.nih.gov/bioproject/, accessed on 19 January 2025), and relevant publications indexed in PubMed (https://pubmed.ncbi.nlm.nih.gov/, accessed on 19 January 2025) and Google Scholar (https://scholar.google.com/, accessed on 19 January 2025) using the same keywords and selection criteria. As a result, 23 relevant experiments were included in the initial dataset for analysis, of which 19 are available through the GEO database: GSE60865 [92], GSE111062 [8], GSE117296 [12], GSE117298 [12], GSE131545 [93], GSE132626 [94], GSE134391 [16], GSE137650 (unpublished data); GSE138196 [13], GSE141916 [14], GSE147962 [15], GSE158898 [53], GSE201015 [95], GSE208563 [9], GSE236872 [18], GSE242435 [19], GSE251796 [17], GSE268410 [96], GSE277977 [97], as well as 4 experiments presented in BioProject: PRJNA650313 [98], PRJNA699408 [99], PRJNA817005 [100], and PRJNA899318 [32]. Detailed information on the final set of experiments included in our analysis is provided in Appendix A.

### 3.2. Preliminary Data Analysis and Quality Control

The initially selected transcriptomic data in sequence read format (.FASTQ) were downloaded using their corresponding SRA identifiers via the SRA Toolkit (https://github.com/ncbi/sra-tools, version 3.1.1, NCBI, Bethesda, MD, USA). The prefetch command was used to retrieve the data, and fasterq-dump –split-files was employed to convert the downloaded sequence read archives, following the official documentation (https://www.ncbi.nlm.nih.gov/sra/docs/sradownload/, accessed on 19 January 2025).

The read alignment was performed using the HISAT2 (https://daehwankimlab.github.io/hisat2/, accessed on 19 January 2025, version 2.2.1, Johns Hopkins University, Baltimore, MD, USA). As a reference, we used the *A. thaliana* unmasked genome assembly (TAIR10, Ensembl Plants release 60, https://plants.ensembl.org/Arabidopsis_thaliana/Info/Index, accessed on 19 January 2025). A reference genome index was built using the hisat2-build command and subsequently used for read alignment, which was carried out for individual transcriptomic libraries using the hisat2 command with default parameters. To reduce disk space usage, the sequence alignment map (SAM) files were converted to binary format (BAM) using samtools view (https://www.htslib.org/, accessed on 19 January, version 1.16.1, Wellcome Sanger Institute, Hinxton, UK).

Quality control of the resulting alignments was performed using RNA-SeQC (https://github.com/getzlab/rnaseqc, version 2.4.2, Broad Institute of MIT and Harvard, Cambridge, MA, USA), and key quality metrics for each experiment and library are provided in Appendix A.

Experiments and replicates that demonstrated acceptable quality metrics (≥85% uniquely mapped reads, exonic rate ≥ 70%, and high-quality reads ≥ 80%) were retained for downstream analysis. In particular, gene-level read counting was performed using the featureCounts (https://subread.sourceforge.net/featureCounts.html, version 2.0.3, WEHI, Parkville, Australia) with default parameters and the TAIR10 genome annotation file in .gtf format. The resulting gene count data for individual libraries were then aggregated into count matrices within each GSE/BioProject experiment using a Python script based on the pandas.concat function (version 2.1.4). The archive containing the resulting count matrices is provided as Appendix A.

### 3.3. Identification of Differentially Expressed Genes in Individual Experimental Conditions and Classification of Transcriptomic Datasets

The resulting read count matrices were used to identify DEGs under individual stress conditions (58 stress-specific groups; see Appendix A). For this purpose, the edgeR package [101] (https://bioconductor.org/packages/release/bioc/html/edgeR.html, accessed on 19 January 2025, version 4.4.1, WEHI, Parkville, Australia), was used within the R (https://www.r-project.org/, accessed on 19 January 2025, version 4.4.1, The R Foundation, Vienna, Austria) statistical environment RStudio Desktop (version 2024.12.1+563, Posit PBC, Boston, MA, USA).

Statistical analysis was performed following the standard edgeR workflow:Read count matrices (from Section 3.2) were loaded from CSV files, including the first column (gene identifiers) and the first row (sample names). An experimental design vector was created, specifying the treatment conditions: control libraries (photosynthetic tissues under non-stress conditions) and stress-induced libraries (photosynthetic tissues exposed to excess light; light intensity ≥ 500 µmol·$m^{-2}\cdot$$s^{-1}$.A DGEList object was created, containing the count matrix and the experimental design vector for downstream analysis.Genes with low expression were filtered out using a threshold of cpm<1 in more than half of the samples.Normalization was performed using the TMM (trimmed mean of M-values) method via the calcNormFactors function.Differential expression analysis was conducted using the exactTest function, comparing control and excess light conditions. False discovery rate (FDR) correction was applied using the Benjamini–Hochberg method. The following thresholds were used to identify DEGs:Upregulated: FDR ≤ 0.05 and log_2_FC ≥ 0.5;Downregulated: FDR ≤ 0.05 and log_2_FC ≤ –0.5.Two comparison strategies were applied depending on the availability of time-matched controls:I For six experiments with time-matched controls (GSE111062, GSE117298, GSE137650, GSE138196, GSE251796, PRJNA817005), statistical comparisons were made between paired time points (e.g., HL stress at 6 h vs. control at 6 h; HL stress at 12 h vs. control at 12 h, etc.).IIFor the remaining 15 experiments with a single control group, all stress-exposed samples were compared against the common control (e.g., HL stress at 6 h vs. control; HL stress at 12 h vs. control, etc.), i.e., all time points of light stress were analyzed relative to the same baseline.

### 3.4. Construction of the AraLightDEGs Knowledge Base for High Light-Responsive DEGs

#### 3.4.1. Structure of the Database

The AraLightDEGs database was implemented using the MariaDB (https://mariadb.org/, accessed on 19 January 2025) relational database management system (version 10.11.11, MariaDB Corporation AB, Espoo, Finland). The enhanced entity–relationship diagram of the database schema is presented in Figure 9.

The central table of the database, expression_data, contains information on DEGs across all experimental conditions. It includes 218,335 records with six fields:expression_id is the primary key; a sequential identifier of each record.condition_id is the identifier of the excess light condition, representing a unique combination of key experimental details (e.g., GSE132626_LEAF_15_500_1_hour).gene_id is the gene identifier in TAIR10 format (e.g., AT1G01010).logFC is the ${\text{log}}_{2}$ fold change in gene expression under excess light conditions relative to the control.logCPM is the average gene expression in ${\text{log}}_{2}$ counts-per-million units across the experiment to which the condition belongs.FDR is the false discovery rate-adjusted *p*-value, calculated using the Benjamini–Hochberg procedure, indicating the statistical significance of the DEG.

The gene_ontology table contains GO annotations for all identified DEGs. The information was obtained from the latest release of the TAIR database (https://www.arabidopsis.org/download/list?dir=Public_Data_Releases%2FTAIR_Data_20231231, accessed on 19 January 2025). This table includes 159,353 records in three fields:gene_index is the primary key; a sequential identifier of the record.gene_id is the gene identifier in TAIR10 format (e.g., AT1G01010).gene_ontology is the GO term associated with the corresponding gene.

For genes that lack GO annotation, the gene_ontology field is left empty. Genes associated with multiple GO terms have multiple entries in this table.

The genes table provides functional descriptions of DEGs and information on TF families. Gene descriptions (field gene_description) were obtained from Ensembl Plants release 60 (https://plants.ensembl.org/Arabidopsis_thaliana/Info/Index, accessed on 19 January 2025), while TF family annotations (field tf_family) were retrieved from Plant-TFDB 4.0 (https://planttfdb.gao-lab.org/download.php, accessed on 19 January 2025) [102]. The tf_family field was filled only for genes annotated as TFs.

The primary key of this table is gene_id, which corresponds to the identically named fields in the expression_data and gene_ontology tables.

The experimental_conditions table contains metadata for all excess light conditions represented in the knowledge base. It includes 58 records across six fields:condition_id is the primary key, identical to the field of the same name in the expression_data table; it represents a meaningful combination of the remaining five fields.experiment_id is the identifier of the corresponding experiment (in GEO or BioProject NCBI format).age_days is plant age in days at the time of sampling.tissue is the tissue type analyzed (either leaf or seedling).intensity_PPFD is the intensity of excess light in µmol $m^{-2}$ $s^{-1}$.duration_min is the duration of excess light exposure in minutes.

The database structure was designed in accordance with data normalization principles to ensure internal consistency and efficient operation, with compliance to the third normal form (3NF) to minimize data redundancy. To enable efficient querying and filtering by gene attributes and experimental conditions, three auxiliary tables (gene_ontology, genes, and experimental_conditions) are linked to the core table of DEGs, expression_data, via foreign keys, and vice versa. These relationships are depicted with arrows in Figure 9.

Specifically, the expression_data and genes tables are connected via the gene_id field, which serves as a foreign key in both tables and corresponds to TAIR-format gene identifiers. Likewise, expression_data and gene_ontology are linked through the shared gene_id field. The experimental_conditions table is related to expression_data through the condition_id field, which acts as a foreign key in both directions.

#### 3.4.2. Implementation of the Server-Side and User Interface Components

The frontend of the knowledge base was implemented using HTML5, JavaScript, and CSS. Specifically, three web pages were developed with the following content:

**“About” page** (https://www.sysbio.ru/aralightdegs/about, accessed on 16 July 2025): This page provides general information about the developed knowledge base, including a brief description of all included experiments and links to the corresponding datasets in the NCBI GEO and BioProject repositories.

**“Search” page** (https://www.sysbio.ru/aralightdegs/search, accessed on 16 July 2025): This page enables users to search for DEGs using SQL-like logic applied to the contents of the knowledge base via a variety of filter options. Search instructions are provided through a dropdown menu interface. The main search action is triggered via the Search in AraLightDEGs button, with two additional quick search buttons available at the top of the page for short- and long-term DEGs, respectively.

Users have access to the following types of filters:Filters by experimental conditions (light intensity and exposure duration, tissue type, and plant age).Filters by DEG identification frequency and expression levels.Filters by TAIR10 gene identifiers, GO terms, or selection of TFs only.

Results can be sorted dynamically based on various table columns. Upon executing a search, two bar plots are rendered, displaying the distribution of DEG identification frequencies for upregulated and downregulated genes. Users can interactively select DEGs of interest from these plots (e.g., those most frequently upregulated under excess light conditions).

For groups of selected DEGs, users can:Query predicted protein–protein interactions via the STRING database v12.0 [103] (https://string-db.org/, accessed on 19 January 2025), implemented via the STRING API (https://string-db.org/help/api/, accessed on 19 January 2025).Download a line-by-line list of DEG identifiers for downstream analysis outside the knowledge base.

The main search output is presented as a table at the bottom of the page, where each row corresponds to a unique DEG that passes the selected filters. For each gene, the following data are provided: functional description, TF family (if applicable), frequency of up- and downregulation across experimental conditions, average ${\text{log}}_{2}$ fold change for up- and downregulation, average light intensity and exposure duration at which the gene is identified as a DEG, and average expression level across relevant conditions.

The search results can be exported as a CSV file. Additional columns can be included in the export, such as the list of associated Gene Ontology terms, the list of experimental conditions where the gene is identified as up- or downregulated, and the list of ${\text{log}}_{2}$ fold changes for each case.

**“Download” page** (https://www.sysbio.ru/aralightdegs/download, accessed on 16 July 2025): This page provides access to three main download options related to the datasets included in the knowledge base. Specifically, users can download DEG lists for all 58 excess light conditions as individual tables. Additionally, count matrices grouped by experiment (21 in total, corresponding to their NCBI GEO/BioProject identifiers) are available for download. A comprehensive table describing all the experiments included in the knowledge base is also provided. This table includes information on plant cultivation conditions, photoperiod duration, light intensity in control and stress groups, light spectrum/type of lamps used to induce excess light conditions, and other experimental parameters. Together, these resources enable users to perform independent re-analyses of the data.

The backend of the knowledge base was developed in Python (version 3.9), using the Flask framework (version 2.3.3) along with supporting libraries: Werkzeug (2.3.7), Flask-Cors (3.0.10), Flask-Limiter (2.4.0), and mysqlclient (2.1.1) to manage SQL query logic from the frontend to the database. In addition, pandas (2.1.4) and numpy (1.26.4) were used to support data processing functionality. The deployment of the knowledge base utilized gunicorn (21.2.0), and for package compatibility and integrity of content on the server, the entire system was containerized using Docker. User request routing is handled via an nginx proxy server.

### 3.5. Development of Transcriptomic Meta-Analysis Pipeline AraLightMeta

The AraLightMeta pipeline was developed to perform a comprehensive meta-analysis of *A. thaliana* DEGs in response to high light conditions in both leaves and seedlings. Its primary objective is to integrate and analyze heterogeneous transcriptomic datasets to identify consistently DEGs, reveal functional enrichment patterns, and reconstruct GRNs associated with the plant’s high light response.

Specifically, the pipeline performs data analysis in three key steps:**Clustering and classification of experimental conditions.** This includes GO analysis of the most frequently identified DEGs in the AraLightDEGs database.**Selection and classification of the most relevant DEGs** associated with short- and long-term high light responses, distinguishing between upregulated and downregulated genes.**Coexpression and machine learning analysis** aimed at reconstructing GRNs associated with specific stress conditions.

The complete version of this pipeline is publicly accessible on GitHub (https://github.com/av-bobrovskikh/AraLightMeta/, accessed on 16 July 2025).

#### 3.5.1. Needed Packages and Input Data Structure

The AraLightMeta pipeline relies on functions from multiple R packages, which must be installed prior to execution. In total, it utilizes 22 packages from the CRAN repository: biomaRt [104], broom, circlize [105], dendextend [106], dplyr, edgeR, ggplot2 [107], ggraph, ggvenn, gridExtra, igraph, networkD3, pheatmap, purrr, RColorBrewer, reshape2, scales, stringr [108], tidygraph, tidyr, tidyverse [109], and WGCNA [110]. Additionally, the pipeline requires three packages from Bioconductor: clusterProfiler [111], org.At.tair.db, and GENIE3 [112].

For permutation testing during the computation of the GENIE3 weight matrix, the test_edges function from the DIANE package is used. Users who wish to perform re-computation of this step must also install its dependencies, which include swfscMisc, rfPermute, HDInterval, kknn, and modeest.

To execute the AraLightMeta pipeline, the following input data files are required:aralightdegs_counts_metadata.csv: This file contains raw count matrices across all experimental conditions (280 in total), along with metadata such as tissue type, light intensity, duration of exposure, and experimental ID. It is utilized for data clustering and classification (Step 1) and for GRN reconstruction (Step 3).aralightdegs_all_degs.csv: A precomputed table of DEGs derived from the AraLightDEGs database, containing gene IDs, descriptions, TF family annotations, up/downregulation statistics, fold changes, and lists of experiments in which up- or downregulation was detected. The table was generated from the AraLightDEGs database (https://www.sysbio.ru/aralightdegs/search, accessed on 16 July 2025) using the following parameters:Remove outlier stress conditions from analysis.Minimum light intensity: 500 µmol·$m^{-2}\cdot$$s^{-1}$.Maximum light intensity: 2000 µmol·$m^{-2}\cdot$$s^{-1}$.Tissue types: Leaves and seedlings.Minimum plant age: 5 days.Maximum plant age: 49 days.Minimum duration of high light treatment: 0 min.Maximum duration of high light treatment: 7200 min.Minimum frequency threshold (total number of DEG identifications across selected experimental conditions): 1.Minimum average expression (in $\log_{2}\mathrm{CPM}$ units): 2.Essentially, these parameters correspond to the following query to the database: Retrieve all available DEGs except those from the two outlier experiments. The result table can be regenerated with alternative parameters if required.The downloadable result table includes the following:Gene descriptions.TF family annotations.Lists of experimental conditions showing upregulation.Lists of experimental conditions showing downregulation.Lists of $\log_{2}\mathrm{FC}$ values for conditions with upregulation.Lists of $\log_{2}\mathrm{FC}$ values for conditions with downregulation.This dataset is used in all stages of the analysis: for GO enrichment of the most frequent DEGs (Step 1), for classifying DEGs into short- and long-term responses in both seedlings and leaves (Step 2), and for adding metadata to GRNs (Step 3).ATH_GO_GOSLIM.txt: A file containing *A. thaliana* GO terms describing biological processes, used for parsing the most relevant DEGs and generating heatmaps in Step 2.Four RDS files containing permutation-tested edges generated by the test_edges function of the DIANE package: edges_test_(long/short)_(leaves/seedlings).rds. These files are derived from GENIE3 weighted matrices computed separately for seedlings and leaves under short- and long-term HL-specific conditions. Inclusion of these precomputed files helps to bypass the computationally intensive step, which otherwise requires at least 128 GB of RAM. However, users can opt to recompute this step within the main pipeline if desired.

#### 3.5.2. First Step of AraLightMeta: Experiments Classification and Frequent Differentially Expressed Genes Analysis

This section corresponds to the initial stage of the analysis block in the AraLightMeta script. It focuses on loading and preprocessing the data, performing hierarchical clustering of unique control and high light stress conditions derived from individual experiments, and aggregating replicate libraries normalized as CPM.

During this step, experimental conditions are further classified into extended groups, and the design of comparisons for subsequent analysis stages is established. For this purpose, metadata describing the characteristics of each dataset (including light treatment duration, light intensity, plant age, plant tissue, and control versus HL conditions) is utilized, alongside the count matrix stored in the file aralightdegs_counts_metadata.csv.

Additionally, for the most frequent DEGs, we computed enrichment for GO biological processes using the clusterProfiler and org.At.tair.db packages. Enrichment was assessed with a significance threshold of *q*-value $\leq 1\times 10^{-5}$ (Benjamini–Hochberg adjustment). Data from the aralightdegs_all_degs.csv table should be used.

Overall, this first step establishes the fundamental framework for subsequent analyses by visualizing the primary characteristics of both the experimental metadata and the DEG datasets.

#### 3.5.3. Second Step of AraLightMeta: Differentially Expressed Genes Classification, Functional Annotation, and Visualization

The second step of the AraLightMeta pipeline focuses on classification of DEGs and a detailed analysis of GO categories specific to HL conditions. From this stage onward, analyses are conducted separately for seedlings and leaves.

Using the AraLightDEGs dataset, the script calculates the frequency of individual DEGs in short-term (from 20 s to 90 min) and long-term (from 2 h to 3 days) HL treatments, based on their regulation patterns in each tissue type. Subsequently, the upper quartile of genes by frequency under each condition is selected and classified into four groups: short-term upregulated, short-term downregulated, long-term upregulated, and long-term downregulated.

These four gene lists are used to identify significantly overrepresented GO biological processes via the clusterProfiler and org.At.tair.db packages, applying the criteria of *q*-value $\leq 1\times 10^{-5}$ and a minimum representation threshold of ≥3% of genes in the target group.

Visualization of intersections among enriched GO terms across these groups is performed using a Sankey diagram (networkD3 package). Additionally, a Venn diagram (ggvenn package) is generated to depict overlaps among the DEG groups.

DEGs falling into two groups simultaneously are classified into combined patterns such as short–long upregulated, short–long downregulated, short upregulated–downregulated, short downregulated–long upregulated, short upregulated–long downregulated, or long upregulated–downregulated. In rare cases, DEGs that overlap in three or more groups are categorized as frequent.

Overall, this stage of the pipeline enables a detailed classification and functional interpretation of DEGs, providing a robust framework for meta-analysis of transcriptomic responses to high light stress.

#### 3.5.4. Third Step of AraLightMeta: Gene Regulatory Networks Reconstruction

The third step of the AraLightMeta pipeline focuses on reconstructing the potential GRNs involved in the HL response of leaves and seedlings. This process begins with coexpression analysis using the classical WGCNA pipeline, implemented via the run_wgcna function.

The input for WGCNA comprises count matrices containing genes classified in specific upregulated and downregulated groups for leaves and seedlings, as determined in the previous analysis step. WGCNA is executed with the following parameters: min_module_size = 30, min_correlation = 0.30 for the leaf dataset, and min_correlation = 0.55 for seedlings. A higher correlation threshold is used for seedlings due to their more consistent coexpression patterns. The soft-thresholding power parameter is selected automatically. The TOMType is set to “unsigned” to identify both positively and negatively correlated gene pairs.

Next, separate count matrices are prepared for the identified coexpressed genes and TFs, which serve as inputs for the GENIE3 machine learning algorithm. GENIE3 predicts potential TF–target gene interactions. From the resulting weight matrices, significant TF–gene edges are filtered using the test_edges function from the DIANE package, with the following parameters: nTrees = 1000, nShuffle = 1000, and density = 0.002 for seedlings and density = 0.065 for leaves.

Subsequently, an intersection of the predicted edges from WGCNA and GENIE3 is performed, retaining only those links confirmed by both methods in the final GRN. Cluster analysis is then applied, with GO enrichment and node meta-annotations generated using the analyze_and_annotate_clusters function.

The final GRNs integrate edges representing varying levels of coexpression strength. Thresholds are adaptively chosen to balance network complexity and biological interpretability. Specifically:For long-term HL response in leaves, the correlation threshold is set to 0.3 (weak coexpression);For short-term HL leaves and long-term HL seedlings, the threshold is set to 0.5 (strong coexpression);For short-term HL seedlings, the threshold is set to 0.6 (strong coexpression).

The resulting networks are visualized via the custom visualize_network function. Visualizations include cluster membership, enriched GO terms, gene type (TF or target), and regulatory direction (upregulated, downregulated, or mixed). The up/down status is assigned based on gene regulation in the specific network context. Genes labeled as “Mixed” represent cases where a small number of genes appear simultaneously in both the upregulated and downregulated subsets for the target group, or fall into the “Frequent” group as defined in Section 3.5.3.

At all stages of the developed pipeline, comprehensive visualizations are produced in the form of high-resolution PDF images (300 dpi), as well as tables in .tsv format. These images were used to prepare the figures in this study and were slightly refined manually prior to publication.

## 4. Conclusions

We conducted the largest meta-analysis to date of *A. thaliana* transcriptomic data under high light stress, focusing on photosynthetic tissues. Our study not only recapitulated the regulation of canonical genes and pathways that respond to HL, but also uncovered novel regulatory interactions specific to long-term adaptation to excessive light in leaves. Using an integrative gene regulatory network reconstruction approach, we identified key TFs orchestrating distinct HL responses: *CRF3* and *PTF1* as regulators of antioxidant–jasmonate crosstalk; *ATWHY2*, *WHY3*, and *emb2746* (newly revealed) in chloroplast–mitochondrial coordination; *AT2G28450* in ribosome biogenesis upregulation; *AT4G12750* in methyltransferase-mediated processes. These TFs coupled with strongly upregulated redox-related DEGs (Section 2.4) could be considered as priority targets for engineering HL-tolerant *A. thaliana* lines. Notably, our meta-analysis demonstrates that synthesizing dozens of experimental conditions with hundreds of replicates can reveal novel interactions in stress GRNs, even in well-studied objects like *A. thaliana*.

Furthermore, we believe that, given the vast amount of publicly available transcriptomic data for *A. thaliana*, similar analysis strategy can be applied to identify core genes and regulators for other stresses. Ultimately, this will facilitate in-depth comparisons of gene regulatory networks involved in different types of stress responses in various tissues. Moreover, promising horizons are opening up in the analysis of single-cell transcriptomic data from plants, as this technology is actively developing [88], and such data will allow us to obtain even more detailed models of gene regulation in individual cell types, which will allow us to refine the results of bulk transcriptome meta-analyses.

As part of this work, we established the information resource AraLightDEGs (https://www.sysbio.ru/aralightdegs/, accessed on 16 July 2025), which can serve as a valuable tool for comparative stress transcriptomics in *A. thaliana*. Furthermore, the entire data analysis and visualization pipeline implemented in R, named AraLightMeta (https://github.com/av-bobrovskikh/AraLightMeta, accessed on 16 July 2025), has been released as open-source software.

## Figures and Tables

**Figure 1 ijms-26-07790-f001:**
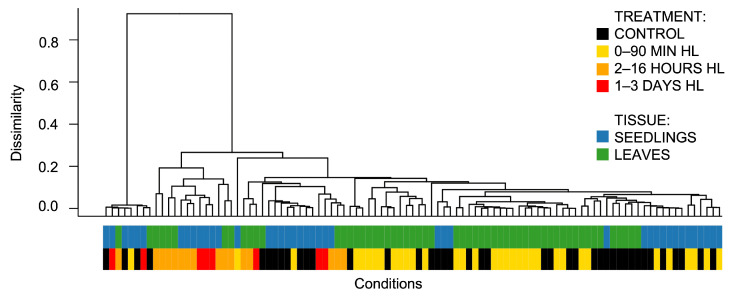
Clustering of transcriptomic libraries from photosynthetic tissues of *A. thaliana*, consisting of 99 conditions derived from 21 high light treatment experiments (41 controls and 58 HL-treated conditions). Control samples are indicated in black, short-duration high light conditions (ranging from 20 s to 90 min) are shown in yellow, medium-duration conditions (2–16 h) are shown in orange, and long-duration conditions (from 1 to 3 days) are shown in red. The far-left cluster represents outliers originating from experiments PRJNA699408 and GSE251796 (comprising 8 conditions). The dendrogram was constructed using UPGMA, based on a dissimilarity matrix calculated as 1−|Pearsoncorrelationmatrix|. The averaged counts-per-million (CPM) expression values for each experimental condition were used to calculate Pearson correlations. A total of 16,000 genes with an average expression of ≥3 CPM were used for this calculation.

**Figure 2 ijms-26-07790-f002:**
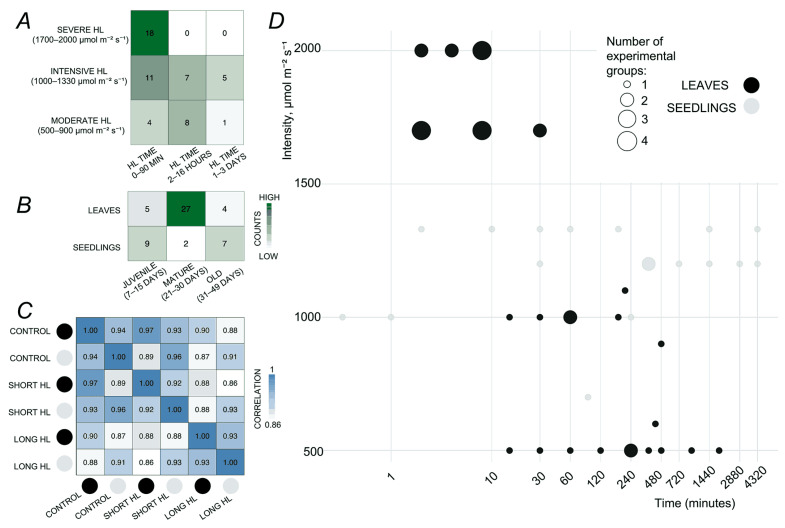
Distribution of characteristics across 54 experimental groups included in the analysis of high light treatments. (**A**) Distribution of experimental groups based on treatment parameters, specifically the duration and intensity of high light exposure. Classification was conducted using three intensity levels (moderate, intensive, and severe high light) and three time intervals (0–90 min, 2–16 h, and 1–3 days). The intensity of green shading reflects the number of replicates within each group. (**B**) Distribution of experimental groups according to plant characteristics, including developmental stage (juvenile, mature, or old) and the type of tissue sampled for transcriptomic analysis (leaves or seedlings). (**C**) Correlation matrix of aggregated transcriptomic data, organized according to the final classification based on exposure duration (short high light, up to 90 min; and long high light, from 2 h to 3 days) and tissue type (seedlings or leaves). Black dots represent groups derived from leaf samples, while gray dots correspond to seedling groups. The intensity of the blue shading indicates the magnitude of the correlation coefficients between groups. (**D**) Distribution of groups in the time–intensity space. Black dots indicate leaf replicates, while gray dots indicate seedling replicates, with the size of each dot proportional to the number of experimental groups represented. The X-axis denotes the duration of light exposure in minutes (displayed on a $\log_{10}$ scale for clarity), and the Y-axis represents light intensity.

**Figure 3 ijms-26-07790-f003:**
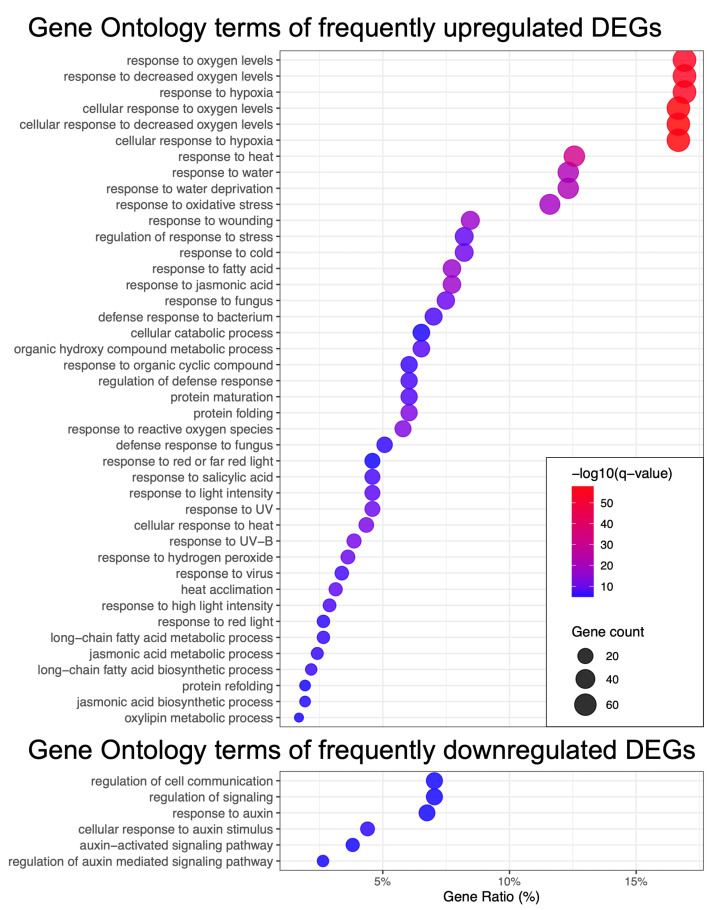
Gene Ontology biological processes significantly associated (*q*-value $<10^{-5}$) with the most frequent differentially expressed genes (upregulated and downregulated) in the *A. thaliana* high light response. Circle size represents the number of genes involved in each process, while red color indicates processes with the lowest *q*-values. Processes are sorted in descending order based on the number of genes.

**Figure 4 ijms-26-07790-f004:**
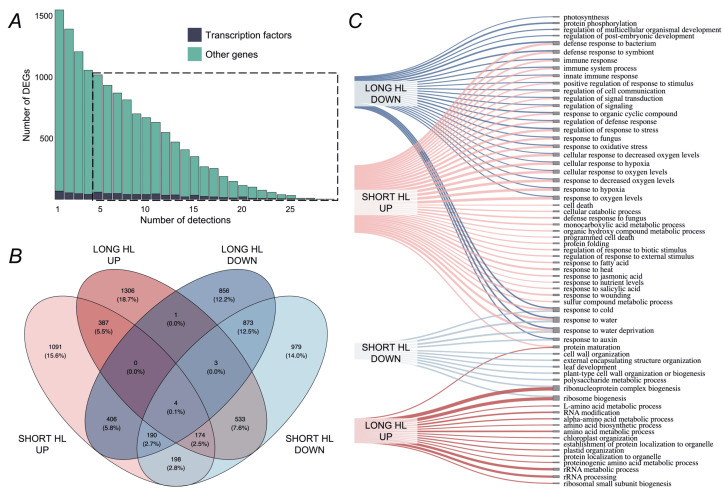
Classification and functional analysis of time-specific DEGs in response to high light in *A. thaliana* leaf transcriptomes. (**A**) Quantitative distribution of DEGs based on their frequency of occurrence. DEGs observed in at least five independent experimental conditions were selected for subsequent analysis and are enclosed within a dashed box. Transcription factors are highlighted in blue, whereas other genes are shown in green. (**B**) Venn diagram illustrating intersections of the most frequently identified DEGs (top quartile) across the upregulated and downregulated subsets within short-term high light (2–60 min) and long-term high light (2–30 h) conditions. Numbers and percentages reflect the quantity of DEGs in each group. In panels (**B**,**C**), upregulated groups are indicated by shades of red, while downregulated groups are indicated by shades of blue. Abbreviations: UP for upregulated DEGs, DOWN for downregulated DEGs. (**C**) Sankey diagram illustrating significantly enriched Gene Ontology biological processes identified across the four DEG groups, applying the following criteria: *q*-value $\leq 1\times 10^{-5}$ and representation of at least 3% of all DEGs within each respective group.

**Figure 5 ijms-26-07790-f005:**
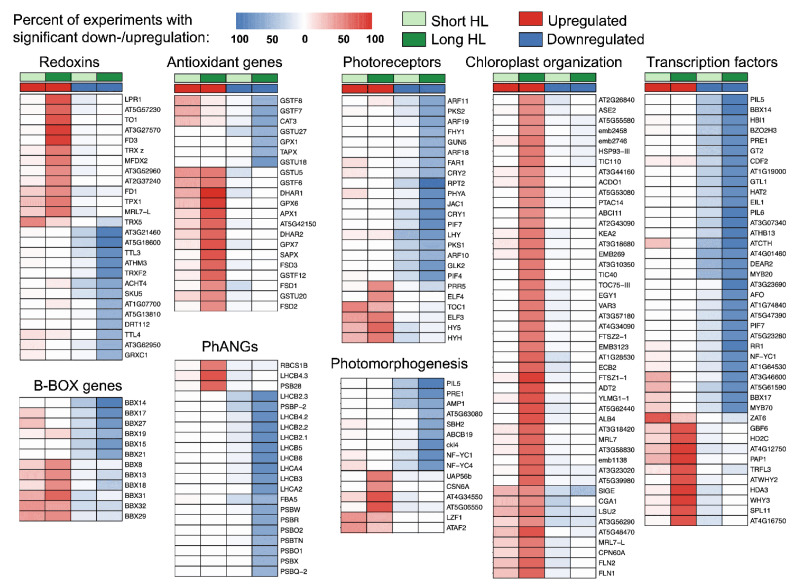
Heatmaps of key pathways and gene classes associated with the high light response in *A. thaliana* leaf transcriptomes. The intensity of shading in each cell is proportional to the percentage of upregulation (indicated in shades of red) or downregulation (indicated in shades of blue) of a given gene across our compiled dataset for short-term and long-term high light responses. From left to right, the columns represent short-term high light upregulated, long-term high light upregulated, short-term high light downregulated, and long-term high light downregulated groups. The abbreviation PhANGs refers to photosynthesis-associated nuclear genes.

**Figure 6 ijms-26-07790-f006:**
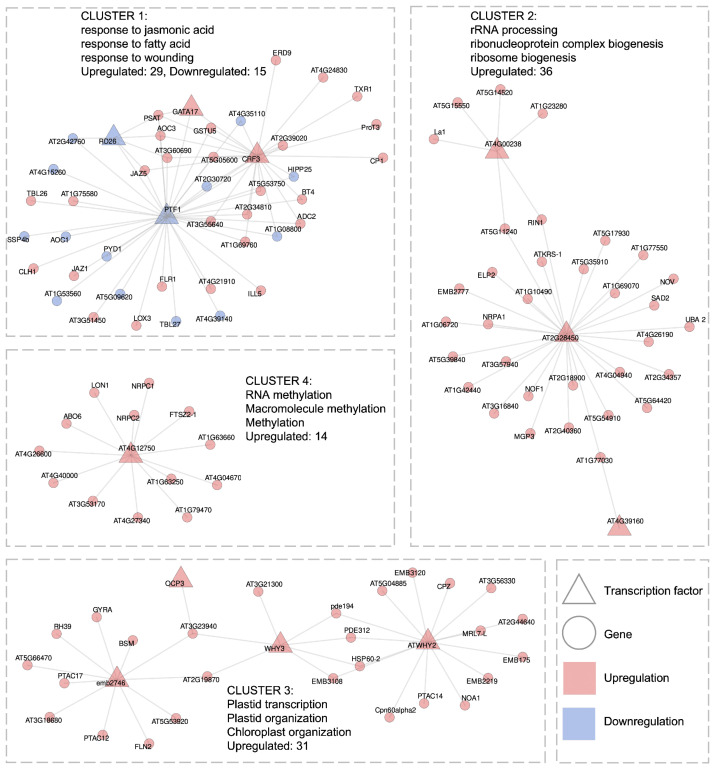
Reconstructed gene regulatory network of the long-term (from 2 to 30 h) high light response in *A. thaliana* (125 nodes and 150 edges). Nodes represent differentially expressed genes, with transcription factors depicted as triangles and other genes as circles. Red nodes indicate upregulated genes, while blue nodes indicate downregulated genes. Clusters are numbered in descending order of their node counts. For each cluster, the top three enriched Gene Ontology terms significantly associated with the genes in the cluster are shown, as determined using clusterProfiler.

**Figure 7 ijms-26-07790-f007:**
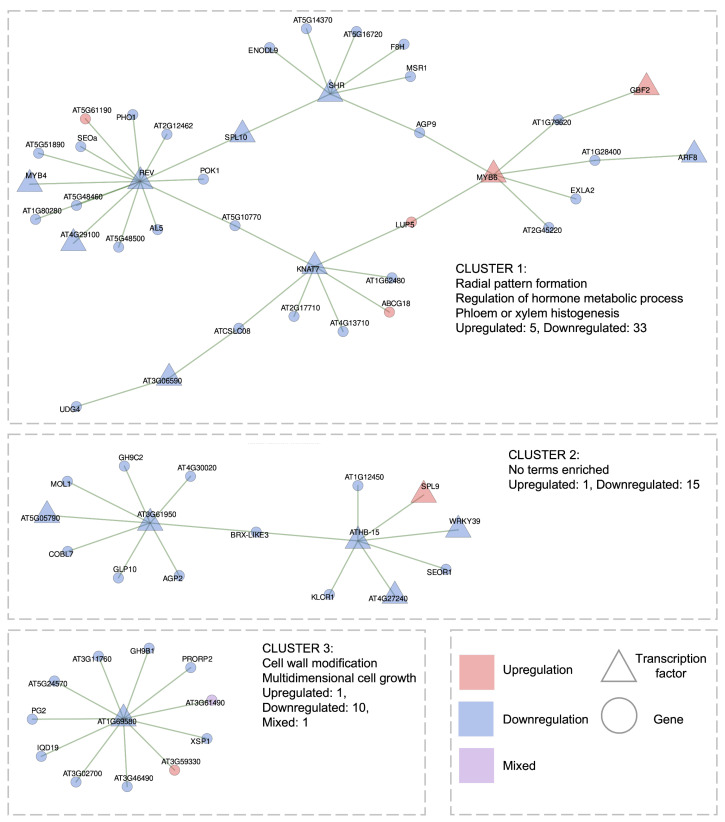
Reconstructed GRN of the short-term (from 2 to 60 min) high light response in *A. thaliana* (66 nodes and 64 edges). Nodes represent differentially expressed genes, where triangles denote transcription factors and circles denote other genes. Upregulated genes are marked in red, downregulated genes are marked in blue, and the gene simultaneously belonging to the frequently upregulated and downregulated groups is highlighted in purple. Clusters are numbered in descending order according to the number of nodes, and for each cluster, and the top three enriched Gene Ontology terms significantly associated with the genes in the cluster are indicated, as determined by clusterProfiler. Gene-to-gene connections are displayed in gray, as all of them fall below the threshold coexpression correlation of 0.5.

**Figure 8 ijms-26-07790-f008:**
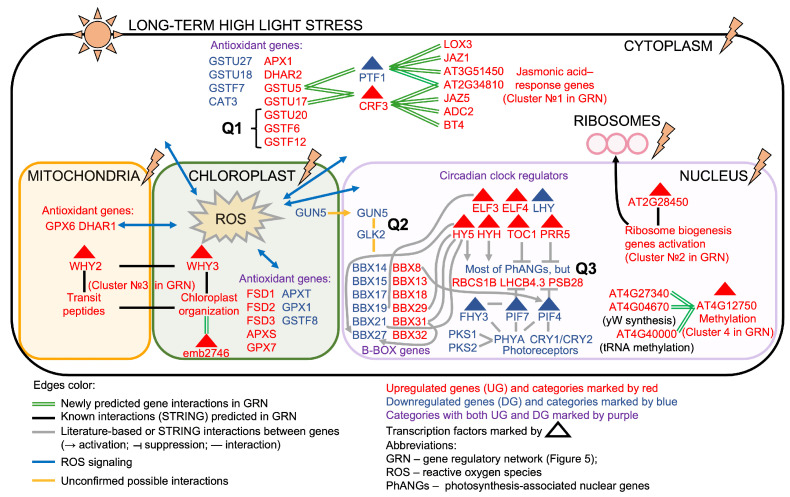
Transcriptional regulation in *A. thaliana* photosynthetic cells under long-term high light. Transcription factors (triangles) and target genes are color-coded by regulation pattern in response to HL (red for upregulated; blue for downregulated). Novel interactions from the reconstructed GRN (see Figure 6) are highlighted in green; known interactions predicted in the GRN are shown in black; known interactions not presented in GRN are shown in gray. Blue arrows indicate signaling pathways associated with reactive oxygen species, and orange arrows/lines represent unconfirmed associations. Open questions (**Q1–Q3**) are addressed in the text below.

**Figure 9 ijms-26-07790-f009:**
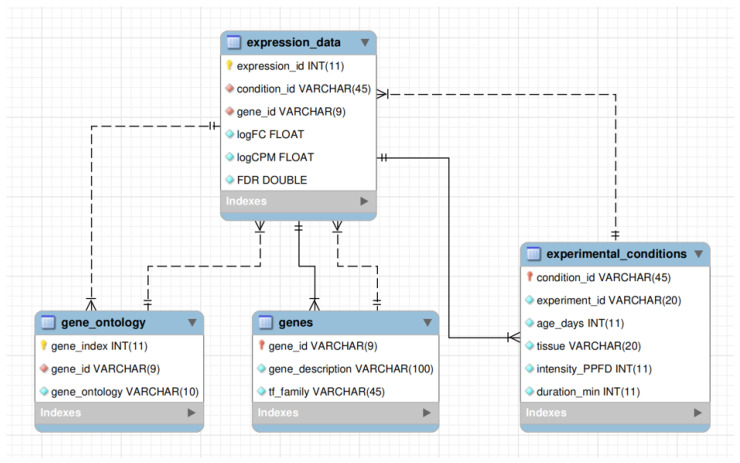
Enhanced entity–relationship diagram of the AraLightDEGs database of differentially expressed genes in *A. thaliana* in response to excess light. The database consists of four tables, with relationships (foreign keys) indicated by directed lines. Each table shows field names and data types. Solid lines represent strong relationships (unique keys in parent tables with possible duplication in child tables’ foreign keys, i.e., one-to-many or one-to-one relationships). Dashed lines indicate weak relationships (parent keys may be duplicated in foreign keys of child tables, i.e., many-to-many or many-to-one relationships).

## Data Availability

Data generated in this research are openly available at the developed resource AraLightDEGs (https://www.sysbio.ru/aralightdegs/, accessed on 16 July 2025); the complete pipeline and results are available on GitHub (https://github.com/av-bobrovskikh/AraLightMeta/, accessed on 16 July 2025). Other additional materials and results can be provided to those interested upon reasonable request.

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
