# Peer review of "Identification of Key Differentially Expressed Genes in *Arabidopsis thaliana* Under Short- and Long-Term High Light Stress"

_ijms, 2025, doi:10.3390/ijms26167790_

Round 1

Reviewer 1 Report

Comments and Suggestions for Authors

Article Identification of Key Differentially Expressed Genes in Arabidopsis thaliana Under Short- and Long-Term High Light Stress
by authors Aleksandr V. Bobrovskikh, Ulyana S. Zubairova, Alexey V. Doroshkov present original data that allow us to evaluate probable mechanisms of expression response to High Light Stress.
One of the problems of this interesting manuscript is that it is not an experimental article, but a review of literature and data from other sources, with a proposal for a methodology for their processing.
For this reason, I think that the publication format should be changed to Review.
Another problem is the theoretical assumption and introduction of a conditional division of objects where expression occurs, at the level of a conditional photosynthetic cell, given at the end of Section 2 - Figure 8.
The authors' concept does not take into account the physical and chemical aspects associated with the structure and localization of the genetic material, there is no detailed analysis of the physical and chemical mechanism of expression regulation caused by High Light Stress.
In this regard, the mechanisms of action of ROS, pH changes, direct impact on structural proteins, nucleic acids, lipid membranes, enzymes are ignored. I recommend that the authors add this to the Discussion.
On the other hand, the authors did not consider the reparation process, which does not allow us to confidently use the concept of stress in this situation. This terminological collapse is of interest at the very beginning of the article, but it remains untouched by the authors. The concept of stress is quite clearly regulated and it is not very correct to call just the impact stress.
After all, it is unclear how exactly the authors determined what exactly was meant.
Obviously, in this context it is hardly possible to correct the work, therefore, in order to avoid embarrassment, I recommend simply discussing three questions:
1. What exactly is High Light Stress for the authors of the works? The authors should not consider works in which recovery was not controlled, which is a methodological error, since if the plants did not recover, then this is not stress, but damage, probably irreversible. There is no point in taking such works into account, they do not correspond to the scientific approach.
2. Determine the ranges of what the authors of the works under consideration call High Light Stress, to ensure the possibility of correct comparison of data.
3. Reflect in the text of the work, in which the authors reliably controlled the physiological state and reparation of systems and control of reversibility.
Also, to improve the work, I recommend discussing the damage scheme at the molecular level, since the organelle level is not directly related to short-term effects.
In addition, the processes of specific and non-specific response to stress, as well as issues of adaptive reaction, should be logically separated.
I would like to see a simple scheme or correlation analysis of comparisons of the reaction to the short and long effect of "High Light Stress", and, if possible, the consequences upon returning to normal.
The work is a good analysis, with high prospects, so I find its careful design important for increasing progress in this area.

Author Response

Dear Reviewer,
We are deeply grateful for your review of our work. It allowed us to improve the quality of the original manuscript.
We have taken your comments into account, please, find below the details.

Comments 1: Article Identification of Key Differentially Expressed Genes in Arabidopsis thaliana Under Short- and Long-Term High Light Stress by authors Aleksandr V. Bobrovskikh, Ulyana S. Zubairova, Alexey V. Doroshkov present original data that allow us to evaluate probable mechanisms of expression response to High Light Stress. One of the problems of this interesting manuscript is that it is not an experimental article, but a review of literature and data from other sources, with a proposal for a methodology for their processing. For this reason, I think that the publication format should be changed to Review.

Response 1: Thank you! Our work is primarily aimed at summarizing the transcriptomic response of plants and identifying new transcription factors and their targets involved in core mechanisms of response to excessive light. In the Results and Discussion section we do discuss in detail the identified core DEGs in the context of key known systems (section 2.2.2.) to facilitate the interpretation of the obtained results, however, we do not stop there and reconstruct gene networks that provide valuable insights especially for the long-term response to high light (sections 2.3 - 2.4).
Thus, we conduct not only a summary of the results of other studies, but a meta-analysis of raw numerical data relevant to studies already published individually, thereby increasing sensitivity and obtaining not just a sum of known factors, but also predicting new, previously unknown genes involved in the response to increased illumination. In particular, our developed bioinformatic pipeline (https://github.com/av-bobrovskikh/AraLightMeta/) allowed us to identify a number of new interactome interactions in gene regulatory networks, which we indicated in the article, and we also created an information resource (https://www.sysbio.ru/aralightdegs/) in which we provided user interface for accessing preprocessed data and extract them according to search filters. Taking all this into account, we believe that our work is primarily an original research paper, and not just a simple review, since it allowed us to obtain new biological knowledge. Nevertheless, we have revised the Results and Discussion and Conclusions sections so that they emphasize the key aspects of our study in the literature context as well as adding overall text fluidity.

Comments 2: Another problem is the theoretical assumption and introduction of a conditional division of objects where expression occurs, at the level of a conditional photosynthetic cell, given at the end of Section 2 - Figure 8. The authors' concept does not take into account the physical and chemical aspects associated with the structure and localization of the genetic material, there is no detailed analysis of the physical and chemical mechanism of expression regulation caused by High Light Stress. In this regard, the mechanisms of action of ROS, pH changes, direct impact on structural proteins, nucleic acids, lipid membranes, enzymes are ignored. I recommend that the authors add this to the Discussion.

Response 2: Thank you! We have expanded the discussion of the limitations of our conceptual diagram  at the end of section 2.4 to avoid ignoring of the mechanisms you listed, with appropriate references. Please, see lines 720-737.

Comments 3: On the other hand, the authors did not consider the reparation process, which does not allow us to confidently use the concept of stress in this situation. This terminological collapse is of interest at the very beginning of the article, but it remains untouched by the authors. The concept of stress is quite clearly regulated and it is not very correct to call just the impact stress. After all, it is unclear how exactly the authors determined what exactly was meant.

Response 3: We appreciate the reviewer’s insightful remark regarding the use of the term "stress" and the overall importance of repair processes in response to HL conditions. Indeed, the concept of stress in plant physiology implies not only the perception of external impact but also the activation of protective and restorative mechanisms, including the repair of damaged components such as photosystem II. We found two photosystem II repair genes to be robustly upregulated under long-term high light stress conditions: AT5G01500 (Thylakoid ADP/ATP Carrier) and PSB28. It can be explained by the primary regulation of PSII repair on post-transcriptional level (https://doi.org/10.1105/tpc.105.037705). Our transcriptomic data reveal a sustained upregulation of genes involved in chloroplast organization and their possible transcription regulators WHY2, WHY3, and emb2746. In addition, a number of antioxidant genes and redoxins are robustly upregulated (shown in Figure 5). Also, in response to your previous comment, we mentioned the photosystem II repair processes in the context of discussing our conceptual scheme (lines 725-726). Thanks again, your comments motivated us to double-check the data. 

Comments 4: Obviously, in this context it is hardly possible to correct the work, therefore, in order to avoid embarrassment, I recommend simply discussing three questions:
1. What exactly is High Light Stress for the authors of the works? The authors should not consider works in which recovery was not controlled, which is a methodological error, since if the plants did not recover, then this is not stress, but damage, probably irreversible. There is no point in taking such works into account, they do not correspond to the scientific approach.
2. Determine the ranges of what the authors of the works under consideration call High Light Stress, to ensure the possibility of correct comparison of data.
3. Reflect in the text of the work, in which the authors reliably controlled the physiological state and reparation of systems and control of reversibility.

Response 4: 
Thank you! Indeed, we initially did not explicitly address the issue of irreversible damage in plants under high light stress. We agree that distinguishing between reversible stress responses and irreversible damage is crucial for the accurate interpretation of physiological and molecular data. Upon careful consideration, we have reviewed the experimental conditions in the studies underpinning our analysis and confirmed that the high light treatments applied — typically lasting from minutes to a few days — do not result in irreversible damage in Arabidopsis thaliana when followed by a recovery period.  
In particular, we have added to the first paragraph of introduction information regarding to all three points to clarify our approach. Please, see lines 34-41.

Comments 5: Also, to improve the work, I recommend discussing the damage scheme at the molecular level, since the organelle level is not directly related to short-term effects. In addition, the processes of specific and non-specific response to stress, as well as issues of adaptive reaction, should be logically separated.

Response 5: Thank you, we have expanded in 2.3.2 a discussion of the molecular mechanisms involved in short-term HL stress and improved sentences clarity with separation of specific and non-specific components. Please, see lines 648-677.

Comments 6: I would like to see a simple scheme or correlation analysis of comparisons of the reaction to the short and long effect of "High Light Stress", and, if possible, the consequences upon returning to normal.

Response 6: Thank you! We did such correlation analysis between the control, short- and long-term high light stress transcriptomes, it discussed in section 2.1.1 in results and depicted at Figure 2. The correlation coefficient between short and long-term HL for leaves is 0.88 (which is relevantly quite low and indicates a significant difference between these time groups of the transcriptomic response and strenghten our classification used), and in the case of seedlings is 0.93 (more coherent response compared to leaves). We did not directly examine the consequences upon returning to normal in our analysis, but according to Huang et al., 2019, "79% of HL72h DEGs restored their expression to the non-stress level"
Your comment motivated us to add this point to discussion of this section. Please, see lines 190-204.

Comments 7: The work is a good analysis, with high prospects, so I find its careful design important for increasing progress in this area.
Response 7: We thank you for your high appreciation of our work. We have taken your comments into account and believe that the article has become better after revision. Additionally, we made many edits to all sections of the results and discussion to increase the fluidity of our text and make it easier to understand.

Reviewer 2 Report

Comments and Suggestions for Authors

In this paper, the author analyzed 21 experiments, covering a total of 58 strong light conditions, generating 218,000 differentially expressed genes (DEGs), corresponding to 19,000 unique genes. It is expected to identify the key genes with differential expression in Arabidopsis thaliana under short-term and long-term strong light stress. Although the author did a lot of literature analysis, a lot of the content still cannot be explained clearly.

  1. In this experiment, how can we ensure that the growth state of the plants is consistent, including the size of the plants, otherwise the experiment cannot accurately reflect the problem.
  2. The author set short-duration strong light conditions (ranging from 20 seconds to 90 minutes) represented by yellow, medium-duration conditions (2 to 16 hours) represented by orange, and long-duration conditions (1 to 3 days). Are there any literature to support these conditions?
  3. In Figure 2, are all the experiments repeated twice?
  4. Does continuous exposure to strong light for three days cause damage to Arabidopsis thaliana, and are there any visible phenotypic changes to the naked eye?.
  5. Did the author identify any specific core genes through this experiment? The entire text provides too many genes, and I didn't see the specific key genes.
  6. The author analyzed a lot of content, but it was very difficult to read and no specific logical thread was found. It is hoped that the author will sort out the sequence carefully.

Author Response

Dear Reviewer,
We are deeply grateful for your review of our work. It allowed us to improve the quality of the original manuscript.
We have taken your comments into account, please, find below the details.

Comments 1: In this paper, the author analyzed 21 experiments, covering a total of 58 strong light conditions, generating 218,000 differentially expressed genes (DEGs), corresponding to 19,000 unique genes. It is expected to identify the key genes with differential expression in Arabidopsis thaliana under short-term and long-term strong light stress. Although the author did a lot of literature analysis, a lot of the content still cannot be explained clearly.

Response 1: 
Thank you! During the revision we improved our manuscript by adding biologically relevant findings in the literature context to the introduction, results and discussion sections. We would just like to emphasize that our paper is not a simple literature review, but we have performed a large-scale uniform bioinformatics meta-analysis starting from raw data, creating the AraLightDEGs resource and the AraLightMeta pipeline for DEG classification and gene regulatory network reconstruction. We explicitly indicate our key findings in the abstract "In particular, long-term HL adaptation involves key TFs such as CRF3 and PTF1 regulating antioxidant and jasmonate signaling; WHY2, WHY3, and emb2746 coordinating chloroplast and mitochondrial gene expression; AT2G28450 governing ribosome biogenesis; and AT4G12750 controlling methyltransferase activity. We integrated these findings into a conceptual scheme illustrating transcriptional regulation and signaling processes in leaf cells responding to long-term HL stress.", in results and disussion (sections 2.3. - 2.4.), in Figure 8, and in conclusions.

Comments 2: In this experiment, how can we ensure that the growth state of the plants is consistent, including the size of the plants, otherwise the experiment cannot accurately reflect the problem.

Response 2: 
Thank you for this important comment. We fully agree that consistency in plant growth stage and developmental status is critical for the accurate interpretation of transcriptomic responses to high light stress.
In this study, our primary goal was to identify robust, conserved transcriptional patterns across diverse published datasets, rather than to analyse separated homogeneous experiments. To address potential variability in plant growth conditions and developmental stages, we took into account several points during our analysis. We have now clarified these four points in the end of revised section 2.1.1 to improve transparency regarding our overall approach. Please, see lines 217-237.

Comments 3:
The author set short-duration strong light conditions (ranging from 20 seconds to 90 minutes) represented by yellow, medium-duration conditions (2 to 16 hours) represented by orange, and long-duration conditions (1 to 3 days). Are there any literature to support these conditions?
Response 3: Thank you for this thoughtful question. Such classification by treatment durations is based on both empirical evidence from the literature and the clustering patterns we observed at Figure 1. Additionally, in the paper of Huang et al. (2019), included in our analysis, the similar classification was used: "...short-term (0.5 h), middle-term (6 to 12 h), and long-term (more than 24 h) HL response patterns..."
To clarify this, we added this part of discussion to the 2.1.1. section. Please, see lines 193-199.

Comments 4:
In Figure 2, are all the experiments repeated twice?
Response 4: Our dataset contains experiments that were performed 1 to 4 times in the same conditions: Tissue - Time - Intensity (Presented in Figure 2 D with various circle sizes proportional to the number of experimental groups in particular conditions).

Comments 5:
Does continuous exposure to strong light for three days cause damage to Arabidopsis thaliana, and are there any visible phenotypic changes to the naked eye?
Response 5: By the data of corresponding article (Huang et al., 2019; GSE111062), where Arabidopsis plants were treated with strong high light (1200 μmol m-2 s-1) for up to three days, leaves showed darkening and pigmentation from 24 hours onwards. However, this did not cause irreversible damage, and after 14 hours of recovery from three days of HL, approximately 79 percent of DEGs had returned to their baseline expression.
To clarify this point and our approach, we have added the relevant information in the introduction. Please, see lines 34-41.

Comments 6:
Did the author identify any specific core genes through this experiment? The entire text provides too many genes, and I didn't see the specific key genes.
Response 6:
Yes, this information is provided in sections 2.3 - 2.4 of the results. We have improved them and the end of introduction to make it clear which sections contain this information. Please, see lines 104-116. Additionally, we improved section 2.3.2 to provide more details on the key genes of the short response to excess light, and discussed the important limitations of our conceptual scheme of the long-term response to HL (Figure 8).

Comments 7:
The author analyzed a lot of content, but it was very difficult to read and no specific logical thread was found. It is hoped that the author will sort out the sequence carefully.

Response 7.
Thank you for understanding the large amount of work we have done. We believe that after the careful revision our article become more easily accessible for readers. In particular, we made many edits to all sections of the results and discussion to increase the fluidity of our text and make it easier to understand.

Round 2

Reviewer 1 Report

Comments and Suggestions for Authors

Article Identification of Key Differentially Expressed Genes in Arabidopsis thaliana Under Short- and Long-Term High Light Stress
by Aleksandr V. Bobrovskikh, Ulyana S. Zubairova, Alexey V. Doroshkov has been supplemented and carefully corrected.
Currently, the manuscript is almost ready for publication.
It remains to correct some minor comments.
Increase the font size of Figure 4
Captions to Figure 8 - Add a bracket and a note that the text is located below "Open questions (Q1—Q3, are addressed in the text."
Tidy up the mentions in the text of "Arabidopsis thaliana", "A. thaliana", "Arabidopsis". What does this mean? First, you need to give the full botanical name of Arabidopsis thaliana (L.) Heynh. at least once.
Second, you should not write simply Arabidopsis - the reason is simple, there are several species of Arabidopsis and not all of them are model plants
Third, uniformity is neatness, it is easier to read: first time the full name, then A. thaliana.
I recommend making a graphic abstract with a code for going to the proposed site.

Author Response

Dear Reviewer,
Thank you again for your thoughtful attention.
We corrected our manuscript according your valuable comments.

Comments 1: Increase the font size of Figure 4.
Reply 1: Thanks, we increased font size and improved overall resolution of this figure for better clarity.

Comments 2: Captions to Figure 8 - Add a bracket and a note that the text is located below "Open questions (Q1—Q3, are addressed in the text."
Reply 2: Thanks, we corrected it: "Open questions (Q1—Q3) are addressed in the text below."

Comments 3:
Tidy up the mentions in the text of "Arabidopsis thaliana", "A. thaliana", "Arabidopsis". What does this mean? First, you need to give the full botanical name of Arabidopsis thaliana (L.) Heynh. at least once.
Second, you should not write simply Arabidopsis - the reason is simple, there are several species of Arabidopsis and not all of them are model plants
Third, uniformity is neatness, it is easier to read: first time the full name, then A. thaliana.

Reply 3: We completely agree and corrected it. The text now uniformly presents the specie name: first time the full name (Arabidopsis thaliana (L.) Heynh.), then A. thaliana.

Comments 4:
I recommend making a graphic abstract with a code for going to the proposed site.
Reply 4:
Thank you for your suggestion. We want to preserve the aesthetics of the graphic abstract as it is now, and besides, we are not sure that the QR code generated by us will work correctly in a few years. In addition, we mention our resource several times in the text of the work, in the abstract and in the data availability statement.

Reviewer 2 Report

Comments and Suggestions for Authors

The author made a lot of revisions according to my suggestions, I have no further opinions.

Author Response

Dear Reviewer,

We thank you for high appreciation of our work.